# LEARNING FROM INTERVAL TARGETS

## ABSTRACT

We consider regression problems where the exact real-valued targets are not directly available; instead, supervision is provided in the form of intervals around the targets—that is, only lower and upper bounds are known. Such a "learning from interval targets" setup arises in domains where labeling costs are high or there is inherent uncertainty in the target values. In these settings, traditional regression loss functions, which require exact target values, cannot be directly applied. To address this challenge, we propose two approaches: (i) modifying the regression loss function to be compatible with interval ground truths, and (ii) formulating a min-max problem where we minimize the typical regression loss with respect to the "worst-case" label within the interval. We provide theoretical guarantees for our methods, analyze their computational efficiency, and evaluate their practical performance on real-world datasets.

## 1 INTRODUCTION

Supervised learning has achieved significant empirical success, largely due to the availability of extensive labeled datasets. However, in many real-world tasks, obtaining target labels is challenging, which hampers the performance of these methods. This difficulty arises either from high labeling costs—for example, certain medical measurements are expensive—or from practical limitations, such as sensors that only record target values at discrete intervals (e.g., every hour), leaving intermediate values unobserved. Prior work has addressed this issue by incorporating additional information into the learning pipeline. For instance, some approaches encourage model outputs to be smooth over unlabeled data (Zhu, 2005; Chapelle et al.), while others enforce models to satisfy constraints derived from domain knowledge, such as physical laws (Willard et al., 2020; Swischuk et al., 2019).

In this work, we focus on regression tasks where only the lower and upper bounds of the target values (intervals) are available. Our setting relates to both weak supervision and learning with side information. Learning with interval targets generalizes supervised learning, which corresponds to the special case where the lower and upper bounds are equal. For many tasks, it is easier and more practical for human labelers to provide interval targets instead of precise single values; thus, these intervals can be viewed as a form of weak supervision. Additionally, in various settings, such intervals are readily available for unlabeled data, either from domain knowledge or inherent properties of the data, serving as side information. A prime example is bond pricing. Unlike stocks, bonds are traded infrequently, so we may observe only a handful of trades over a given time span, resulting in limited labeled data for bond prices. However, numerous bond quotes are available: the bid and ask quotes represent the prices at which dealers are willing to buy and sell, respectively. These bid and ask quotes can be treated as lower and upper bounds of the true bond prices when actual trade prices are not observed.

A natural learning strategy for learning from interval targets is to learn a hypothesis whose outputs always lie within the provided intervals. Despite its simplicity, previous work (Cheng et al., 2023a) has shown that this method leads to a hypothesis that converges to the optimal one under two assumptions: (i) the true target function belongs to the hypothesis class, and (ii) the intervals have an ambiguity degree smaller than 1 (Section 2). However, these assumptions are unlikely to hold in practice. In particular, (ii) is often violated; for example, even in the simple case where the interval is a ball of radius $\epsilon$ around the target value $y$, the ambiguity degree equals 1. It is important to understand whether this approach can be effective under more relaxed assumptions. Our first contribution is a novel theoretical analysis of learning a hypothesis that lies within the target intervals. Our error bound, based on the Lipschitz constant of the hypothesis class, is applicable even when

the ambiguity degree is 1 (Theorem 3.3) and can be extended to the agnostic setting where the target function may not belong to our hypothesis class (Theorem 3.8). Unlike prior bounds (Cheng et al., 2023a) that only consider the asymptotic case as the number of data points $n \to \infty$, our bound is non-asymptotic. The key insight is that, when the hypothesis class is smooth, the outputs for two close inputs cannot differ significantly. As a result, portions of the original intervals can be ruled out, leading to much smaller valid intervals (Figure 1b).

In our second contribution, we explore an alternative approach by learning a hypothesis that minimizes the loss with respect to the worst-case labels within the given intervals. Since we assume that the true target values lie within these intervals, the worst-case loss serves as an upper bound on the regression loss. We consider two variants of the second approach: i) we allow the worst-case labels to be any points within the intervals, ii) we restrict the worst-case labels to be outputs of some hypothesis in our hypothesis class, thereby incorporating the smoothness property. We show that there exists a distribution where the second variant can perform arbitrarily better than the first (Proposition 4.4), indicating that constraining the worst-case labels to the hypothesis class is preferable in the worst-case scenario. We demonstrate the effectiveness of both methods on real-world datasets.

### 1.1 RELATED WORK

Our problem is closely related to partial-label learning, where each training point is associated with a set of candidate labels instead of a single target label (Cour et al., 2011; Ishida et al., 2017; Feng et al., 2020a; Ishida et al., 2019; Yu et al., 2018). In classification with finite label sets, a popular method is to minimize the average loss over the label set (Jin & Ghahramani, 2002), leading to various extensions (Zhang et al., 2017; Wang et al., 2019; Xu et al., 2021; Wu et al., 2022; Gong et al., 2022). Another important approach focuses on identifying the true label from the candidate set (Lv et al., 2020; Zhang et al., 2016; Yu & Zhang, 2016). On the theoretical side, prior work has studied learnability conditions (Liu & Dietterich, 2014; Cour et al., 2011) and developed statistically consistent estimators (Lv et al., 2020; Feng et al., 2020b; Wen et al., 2021) based on the small ambiguity degree assumption or specific label set generating distributions.

In regression, there has been less prior work. Cheng et al. (2023b) considers partial-label regression with a finite label set and Cheng et al. (2023a) later extends it to the interval setting with infinitely many labels. Both works provide statistically consistent estimators, but their theoretical results heavily depend on the small ambiguity degree assumption. We note that the ambiguity degree, first proposed for classification tasks in Cour et al. (2011), may not be suitable for regression tasks. In classification, a hypothesis is either correct or incorrect, and a small ambiguity degree ensures that, with enough observed label sets, we can recover the true label. However, in regression, we are often satisfied with predictions that are sufficiently close to the target—for example, within an error tolerance of $\epsilon$—making the concept of ambiguity degree less applicable.

In our work, we study a projection loss, which is equivalent to the partial-label learning loss (PLL loss) in Lv et al. (2020), and can be seen as a generalized version of the limiting method in Cheng et al. (2023a). We provide a non-asymptotic error bound that does not rely on the ambiguity degree and extend our analysis to the agnostic setting. We provide additional related work in Appendix C.

### 1.2 PRELIMINARIES AND NOTATION

Let $\mathcal{X}$ be the feature space and $\mathcal{Y}$ be the label space. Let $f^*: \mathcal{X} \to \mathcal{Y}$ denote the target function. We use uppercase letters (e.g., $X$) to represent random variables and lowercase letters (e.g., $x$) for deterministic variables. We consider a regression problem where our goal is to learn a function $f: \mathcal{X} \to \mathcal{Y}$ from a hypothesis class $\mathcal{F}$ that approximates the target function $f^*$ in the deterministic label setting. Let $\mathcal{D}$ be the distribution over $\mathcal{X} \times \mathcal{Y}$ where, for each $x \in \mathcal{X}$, the label $y$ is deterministically given by $y = f^*(x)$ and let $p$ be the pdf. **Our goal is to learn a function $f$ that minimizes the expected loss** $\mathrm{err}(f) := \mathbb{E}_{(X,Y) \sim \mathcal{D}} \left[ \ell\big(f(X), Y\big) \right]$ **for some loss function** $\ell: \mathcal{Y} \times \mathcal{Y} \to \mathbb{R}$.

**Interval targets.** However, we assume that we have access only to interval samples of the form $\{(x_i, l_i, u_i)\}_{i=1}^n$, where $l_i$ and $u_i$ are the lower and upper bounds of $y_i$, respectively. While we assume that the label is fixed to $f^*(x_i)$, we allow the intervals—that is, the bounds $(l_i, u_i)$—to be random and assume that each tuple $(x_i, l_i, u_i)$ is sampled from some distribution $\mathcal{D}_I$. In this setting, we aim to explore learning strategies and determine what kinds of guarantees are possible.

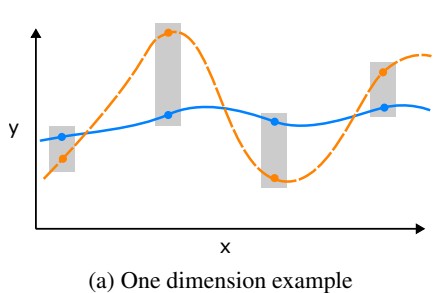 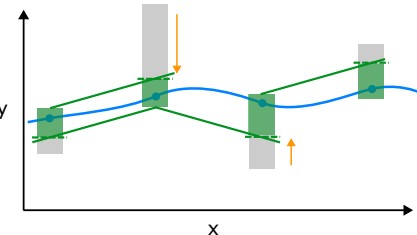

(a) One dimension example

(b) Smooth hypothesis leads to a smaller interval

Figure 1: (1a) An example of learning from intervals where the input is one dimension. The intervals are shown as gray boxes. A natural method is to learn a hypothesis that always lies within these intervals. Here, we illustrate two such hypotheses that are both valid but have different levels of smoothness. (1b) When the hypothesis is smooth (blue line), it lies within intervals much smaller than the original ones, depicted by the green region (Proposition 3.2). We can extend this result to hypotheses that approximately lie within the intervals (Theorem 3.3).

## 2 LEARNING FROM INTERVALS USING A PROJECTION LOSS

Since the target label $y$ always lies within the interval $[l, u]$, a natural strategy is to learn a hypothesis $f \in \mathcal{F}$ such that $f(x) \in [l, u]$ for all $x \in \mathcal{X}$ (Figure 1a). In previous work, Cheng et al. (2023a) analyzed the following strategy:

$$\text{Learn } f \text{ that minimizes the empirical risk of the 0-1 loss: } \sum_{i=1}^{n} \ell_{0-1}(f(x_i), l_i, u_i), \quad (1)$$

where $\ell_{0-1}(f(x), l, u) := 1[f(x) < l] + 1[f(x) > u]$. Using $\ell_1$ loss as the surrogate (equation (12)), they showed that $f$ converges to $f^*$ as $n \to \infty$ if two assumptions are satisfied, (i) Realizability, that is, $f^* \in \mathcal{F}$, (ii) Ambiguity degree is smaller than 1. Ambiguity degree is the maximum probability of a specific incorrect target $y'$, belonging to the same interval $[l, u]$ as the true target $y$:

$$\text{Ambiguity degree}(\mathcal{D}, \mathcal{D}_I) := \sup_{(x,y)\sim\mathcal{D},(x,l,u)\sim\mathcal{D}_I, y'\in\mathcal{Y}, y'\neq y} \Pr(y' \in [L, U]) < 1. \quad (2)$$

These assumptions can be impractical and restrictive. First, our hypothesis class may not contain $f^*$. Second, an ambiguity degree smaller than 1 implies that for any fixed $x$, if we keep sampling the interval $[l, u]$, the intersection of such intervals (in the limit) would only be the set of the true target $\{y\}$; that is, we can recover the true $y$ given an infinite number of intervals. However, this assumption is unlikely to hold in practice because there is usually a gap between the upper and lower bounds and the target $y$. For example, in the simple case where $[l, u] = [y - \epsilon, y + \epsilon]$ (a ball with radius $\epsilon > 0$ around the true target $y$), the assumption fails since $y + \epsilon/2$ always lies within the interval at the same time with the true $y$. Finally, we note that the previous result is an asymptotic bound that only applies when $n \to \infty$. In this work, we relax these assumptions and provide a non-asymptotic generalization bound.

We begin by defining a suitable learning objective. Since the 0-1 loss above is not continuous, it is not suitable for gradient-based optimization techniques. To address this, we relax the loss by considering a projection

$$\pi_\ell(f(x), l, u) := \min_{\tilde{y}\in[l,u]} \ell(f(x), \tilde{y}) \quad (3)$$

for any surrogate loss function $\ell : \mathcal{Y} \times \mathcal{Y} \to \mathbb{R}$. We make the following assumption about $\ell$,

**Assumption 1.** *The loss function $\ell : \mathcal{Y} \times \mathcal{Y} \to \mathbb{R}$ can be written as $\ell(y, y') = \psi(|y - y'|)$ for some non-decreasing function $\psi$, and satisfies $\ell(y, y') = 0$ if and only if $y = y'$.*

The following proposition shows that $\pi_\ell$ is a meaningful proxy for the 0-1 loss, and can be evaluated efficiently by only considering the boundaries of the interval.

**Proposition 2.1.** *Suppose that $\ell : \mathcal{Y} \times \mathcal{Y} \to \mathbb{R}$ is a loss function that satisfies Assumption 1 then $\pi_\ell(f(x), l, u) = 0$ if and only if $f(x) \in [l, u]$, and we can write*

$$\pi_\ell(f(x), l, u) = 1[f(x) < l]\ell(f(x), l) + 1[f(x) > u]\ell(f(x), u). \quad (4)$$

The proof is provided in Appendix D.1. In the rest of the paper, we refer to $\pi_l$ as the **projection loss**. Consequently, the informal goal given in (1) can be formalized as the following objective:

$$\min_f \sum_{i=1}^n 1[f(x_i) < l_i]\ell(f(x_i), l_i) + 1[f(x_i) > u_i]\ell(f(x_i), u_i). \tag{5}$$

## 3 THEORETICAL ANALYSIS OF THE PROJECTION APPROACH

Denote $\widetilde{\mathcal{F}}_\eta := \{f \in \mathcal{F} \mid \mathbb{E}[\pi_\ell(f(X), L, U)] \leq \eta\}$ as a class of hypotheses for which the expected projection loss is smaller than $\eta$. When $\eta = 0$, we have $\widetilde{\mathcal{F}}_0 = \{f \in \mathcal{F} \mid \Pr(f(X) \in [L, U]) = 1\}$ which is a class of hypothesis that always lie within the interval, a property that we aim to achieve for our hypothesis $f$. However, since we only have access to a finite number of data points, we can only hope to learn $f \in \widetilde{\mathcal{F}}_\eta$ for some small $\eta$. Specifically, by a standard uniform convergence argument (e.g. Mohri (2018)), we have that with probability at least $1 - \delta$ over the draws $(x_i, l_i, u_i) \sim \mathcal{D}_I$,

$$\text{for all } f \in \mathcal{F}, \mathbb{E}[\pi_\ell(f(X), L, U)] \leq \frac{1}{n}\sum_{i=1}^n \pi_\ell(f(x_i), l_i, u_i) + 2R_n(\Pi(\mathcal{F})) + M\sqrt{\frac{\ln(1/\delta)}{n}}. \tag{6}$$

Here, $R_n(\Pi(\mathcal{F}))$ is the Rademacher complexity of the function class $\Pi(\mathcal{F}) := \{\pi_\ell(f(x), l, u) \mapsto \mathbb{R} \mid f \in \mathcal{F}\}$ and we assume that the $\pi_\ell$ is uniformly bounded by $M$. Thus, given $n$, $M$, and the empirical loss on observed data (first term in R.H.S.), we have an **upper bound** of $\eta$ which $f \in \widetilde{\mathcal{F}}_\eta$.

**Plan of analysis.** In the rest of this section, we show general bounds for any $f \in \widetilde{\mathcal{F}}_\eta$. In Theorem 3.3 we show that for any $x$, $f(x)$ belongs to a small interval that depends on $M$ and $\eta$. This leads to our main result: a generalization bound on the loss of $f$ with respect to actual labels $y$, thus showing that regression can be done using interval targets (Section 3.3).

### 3.1 EFFECT OF REALIZABILITY AND SMALL AMBIGUITY DEGREE ASSUMPTIONS ON $\widetilde{\mathcal{F}}_\eta$

We begin by examining the implications of the assumptions made in prior work (Section 2). The realizability assumption can be restated as $f^* \in \widetilde{\mathcal{F}}_0$ since the projection loss of $f^*$ is always zero. Second, the small ambiguity degree assumption implies that, for any $x$, the intersection of the intervals can only be the singleton set $\{y\}$. As a result, we have $\widetilde{\mathcal{F}}_0 = \{f \in \mathcal{F} \mid \text{err}(f) = 0\} \neq \emptyset$.

With these assumptions, we can show that minimizing the projection objective will converge to a hypothesis with zero error. The following informal argument summarizes the more elaborate analysis of Cheng et al. (2023b). Here is the high-level idea: let $f_n$ be the hypothesis that minimizes the empirical projection objective (5). Realizability implies that there exists $f^* \in \mathcal{F}$ with an expected loss of zero. Since $f_n$ achieves the empirical risk no larger than that of $f^*$, it must achieve an empirical risk of zero. From equation 6, we have $f_n \in \widetilde{\mathcal{F}}_{\eta_n}$ with high probability, where $\eta_n = 2R_n(\Pi(\mathcal{F})) + M\sqrt{\frac{\ln(1/\delta)}{n}}$. In general, $\eta_n = O(1/\sqrt{n})$, and as $n \to \infty$, we have $\eta_n \to 0$ which means that $\widetilde{\mathcal{F}}_{\eta_n} \to \widetilde{\mathcal{F}}_0$. Consequently, $\text{err}(f_n) \to 0$ since any member of $\widetilde{\mathcal{F}}_0$ has zero error.

However, when the realizability and ambiguity degree assumptions do not hold, there may be $f \in \widetilde{\mathcal{F}}_0$ with $\text{err}(f) > 0$. Additionally, with a finite amount of data, we can only learn a hypothesis $f \in \widetilde{\mathcal{F}}_{\eta_n}$ for some $\eta_n > 0$. In the next section, we will analyze $\widetilde{\mathcal{F}}_\eta$ without relying on the small ambiguity degree assumption and in finite samples. Further, we also provide a formal argument that relates $\eta$ with $n$ and provide an error bound depends on $n$ in Appendix A.

### 3.2 PROPERTIES OF $\widetilde{\mathcal{F}}_\eta$

Although our results extend to the probabilistic interval setting, where multiple intervals $[l, u]$ are drawn for each $x$, we focus on the deterministic interval setting for simplicity. In this case, each $x$ is associated with a fixed interval $[l_x, u_x]$. For a detailed discussion of the probabilistic interval setting, please refer to Appendix E. Now, consider the following characterization of $\widetilde{\mathcal{F}}_0$.

**Proposition 3.1.** *For any $f \in \widetilde{\mathcal{F}}_0$, and $\ell$ that satisfies Assumption 1, we have $f(x) \in [l_x, u_x]$.*

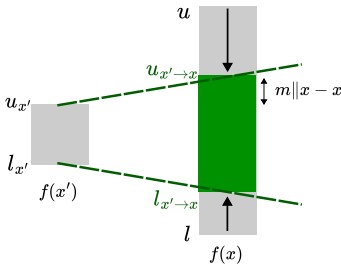 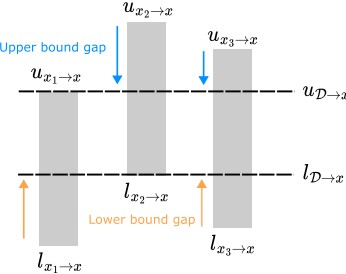

(a) An interval of $x$ induced by $x'$      (b) Upper and lower bound gaps

Figure 2: (2a) Based on the smoothness property, the difference between $f(x)$ and $f(x')$ cannot exceed $m\|x - x'\|$. As a result, the upper and lower bounds of $f(x')$ imply the corresponding bounds for $f(x)$. (2b) The lower bound gap of $x'$ to $x$ is defined as the difference between the lower bound of $f(x)$ induced by $x'$ and the largest lower bound ($\tilde{l}_{\mathcal{D} \to x}^{(m)}$); similarly for the upper bound gap. These gaps are crucial in bounding the size of $r_\eta(x)$ and $s_\eta(x)$ (how much we have to compensate when $f \in \widetilde{\mathcal{F}}_\eta$) where larger gaps lead to larger values (Theorem 3.3).

We will show that the interval in which $f(x)$ must lie can be shrunk (made smaller than $[l_x, u_x]$) if we assume that the class $\mathcal{F}$ contains only $m$-Lipschitz functions. That is, for any $f \in \mathcal{F}$ and any $x, x' \in \mathcal{X}$, the condition $|f(x) - f(x')| \leq m\|x - x'\|$ holds ($L_1$ norm). We define for any $x, x'$,

$$l_{x' \to x}^{(m)} := l_{x'} - m\|x - x'\|, u_{x' \to x}^{(m)} := u_{x'} + m\|x - x'\|.$$

**Proposition 3.2.** *Let $\mathcal{F}$ be a class of hypotheses that are $m$-Lipschitz and suppose that $\ell$ satisfies Assumption 1. Then for any $f \in \widetilde{\mathcal{F}}_0$ and for each $x$ with $p(x) > 0$,*

$$f(x) \in \bigcap_{x'} [l_{x' \to x}^{(m)}, u_{x' \to x}^{(m)}] =: [l_{\mathcal{D} \to x}^{(m)}, u_{\mathcal{D} \to x}^{(m)}]. \tag{7}$$

*Proof.* (Sketch) For $f \in \widetilde{\mathcal{F}}_0$, by Lipschitzness, for any $x, x' \in \mathcal{X}$, we have $|f(x) - f(x')| \leq m\|x - x'\|$ which implies $f(x') - m\|x - x'\| \leq f(x) \leq f(x') + m\|x - x'\|$. We can then replace $f(x')$ with its lower and upper bound, $l_{x'}, u_{x'}$, respectively to achieve the result. □

***Interpretation.*** Note that $l_{x' \to x}^{(m)}$ and $u_{x' \to x}^{(m)}$ provide lower bound and upper bounds of $f(x)$ induced by $x'$, derived from the Lipschitz property of $f$ (Figure 2a). We denote the intersection of all such intervals $[l_{x' \to x}^{(m)}, u_{x' \to x}^{(m)}]$ over all $x'$ by $[l_{\mathcal{D} \to x}^{(m)}, u_{\mathcal{D} \to x}^{(m)}]$. First, we observe that $[l_{\mathcal{D} \to x}^{(m)}, u_{\mathcal{D} \to x}^{(m)}]$ is always smaller than $[l_x, u_x]$ because when we set $x' = x$, we have $[l_{x' \to x}^{(m)}, u_{x' \to x}^{(m)}] = [l_x, u_x]$. Second, as the hypothesis becomes more smooth, the interval $[l_{\mathcal{D} \to x}^{(m)}, u_{\mathcal{D} \to x}^{(m)}]$ gets smaller. To see this, $f$ is more smooth when the Lipschitz constant $m$ is smaller. This implies that $l_{x' \to x}^{(m)} = l_{x'} - m\|x - x'\|$ is larger and $u_{x' \to x}^{(m)} = u_{x'} + m\|x - x'\|$ is smaller. As a result, we have a smaller interval $[l_{x' \to x}^{(m)}, u_{x' \to x}^{(m)}]$ for each $x'$ and which implies a smaller $[l_{\mathcal{D} \to x}^{(m)}, u_{\mathcal{D} \to x}^{(m)}]$. This phenomenon can also be interpreted as implicitly "denoising" the original intervals by leveraging the smoothness of the hypothesis class.

Next, we extend Proposition 3.2 to $\widetilde{\mathcal{F}}_\eta$. The technical challenge is that for $f \in \widetilde{\mathcal{F}}_\eta$, $f(x)$ may lie outside the interval so we can't simply use $[l_{x'}, u_{x'}]$ as lower and upper bounds of $f(x')$. This complicates the application of the Lipschitz property because $f(x')$ can now be arbitrarily large or small for any $x'$, as long as the expected projection loss is smaller than $\eta$. The following result uses a new notion of a bound gap of $f(x)$ induced by $x'$ which is the difference between the lower and upper bounds induced by a given $x'$ and the best lower and upper bounds from all $x'$ (Figure 2b). Formally, the lower bound gap for $f(x)$ induced by $x'$ is defined as $lg_{x' \to x}^{(m)} = l_{\mathcal{D} \to x} - l_{x' \to x}^{(m)}$ while the upper bound gap is defined as $ug_{x' \to x}^{(m)} = u_{x' \to x}^{(m)} - u_{\mathcal{D} \to x}$.

**Theorem 3.3.** *Let $\mathcal{F}$ be a class of functions that are $m$-Lipschitz, and $\ell(y, y') = |y - y'|^p$ for any $p \geq 1$. For any $f \in \widetilde{\mathcal{F}}_\eta$ and for each $x$ with $p(x) > 0$ we have,*

$$f(x) \in [l_{\mathcal{D} \to x}^{(m)} - r_\eta(x), u_{\mathcal{D} \to x}^{(m)} + s_\eta(x)], \text{ where,} \tag{8}$$

$$r_\eta(x) = r \quad s.t. \quad \mathbb{E}_X[(r - lg_{X \to x}^{(m)})_+^p] = \eta, \tag{9}$$

$$s_\eta(x) = s \quad s.t. \quad \mathbb{E}_X[(s - ug_{X \to x}^{(m)})_+^p] = \eta. \tag{10}$$

*Proof.* (Sketch) Our idea is based on the smoothness property of $f$. We can show that whenever $f(x)$ is far from the interval of $[l_{\mathcal{D}\to x}^{(m)}, u_{\mathcal{D}\to x}^{(m)}]$, $f(x')$ is also far away from $[l_{x'}, u_{x'}]$. However, this cannot happen frequently because the expected projection loss is smaller than $\eta$. $\qquad\square$

***Interpretation.*** We compensate for $f \in \widetilde{\mathcal{F}}_\eta$ by adding a buffer of size $r$ and $s$ to the interval derived in Proposition 3.2. If the average lower and upper bound gap is large, then we would have a larger compensation $r, s$. When $\eta = 0$, we have $r = s = 0$. In general, we can bound $r, s$ in terms of $\eta^{1/p}$.

**Proposition 3.4.** *Under the conditions of Theorem* 3.3*, we can bound $r_\eta(x)$ and $s_\eta(x)$, as*

$$r_\eta(x) \leq \inf_\delta \delta + (\eta/\Pr(lg_{X\to x}^{(m)} \leq \delta))^{1/p} \quad and \quad s_\eta(x) \leq \inf_\delta \delta + (\eta/\Pr(ug_{X\to x}^{(m)} \leq \delta))^{1/p}. \quad (11)$$

### 3.3 GENERALIZATION BOUNDS

In the previous section, we showed that the applicable interval of $f(x)$ is smaller than the original interval $[l_x, u_x]$ when the hypothesis class $\mathcal{F}$ is smooth. We can leverage this property to provide a generalization bound for any hypothesis $f \in \widetilde{\mathcal{F}}_\eta$. We denote the reduced interval from Theorem 3.3 as $I_\eta(x) := [l_{\mathcal{D}\to x}^{(m)} - r_\eta(x), u_{\mathcal{D}\to x}^{(m)} + s_\eta(x)]$. We will use the property that for intervals $I_1 = [l_1, u_1], I_2 = [l_2, u_2]$, and for any $y_1 \in I_1, y_2 \in I_2$, and any $\ell$ that satisfies Assumption 1,

$$\ell(y_1, y_2) \leq \max(\ell(l_1, u_2), \ell(u_1, l_2)) =: d(\ell, I_1, I_2). \quad (12)$$

#### 3.3.1 REALIZABLE SETTING

**Theorem 3.5** (Error bound, Realizable setting)**.** *Let $\mathcal{F}$ be a class of functions that are $m$-Lipschitz, assume that $f^* \in \widetilde{\mathcal{F}}_0$, then for any $f \in \widetilde{\mathcal{F}}_\eta$,*

$$\mathrm{err}(f) \leq \mathbb{E}[d(\ell, I_0(X), I_\eta(X))]. \quad (13)$$

Notably, when we minimize the projection objective, $f$ belongs to $\widetilde{\mathcal{F}}_\eta$ with a small $\eta$, where $\eta = O(1/\sqrt{n})$ (Section 3.1). While this bound is straightforward, we remark that it can be tight for certain hypothesis classes. For example, consider the case where $\mathcal{F}$ consists of constant hypotheses and let $n \to \infty$. In this scenario, we have $r_\eta(x) \to r_0(x) = 0$ and $I_\eta(x) \to I_0(x)$. For each $x$, the error bound is given by

$$d(\ell, I_0(x), I_0(x)) = \ell(l_{\mathcal{D}\to x}^{(m)}, u_{\mathcal{D}\to x}^{(m)}) = \ell(\sup_{x'} l_{x'}, \inf_{x'} u_{x'}), \quad (14)$$

representing the loss between the boundaries of the intersected intervals. It is tight since the inequality holds when $f^*$ and $f$ each take values at the respective boundaries of the intersected interval.

#### 3.3.2 AGNOSTIC SETTING

Now, we study the agnostic setting, where we do not assume the existence of such $f^*$ in $\mathcal{F}$. Instead, we focus on comparing with $f_{\mathrm{OPT}} = \arg\min_{f \in \mathcal{F}} \mathrm{err}(f)$, the hypothesis in $\mathcal{F}$ with the smallest expected error. First, we show that, in contrast to the realizable setting, simply learning a hypothesis $f \in \mathcal{F}$ that always lies within the interval by minimizing the projection loss may not converge to $f_{\mathrm{OPT}}$. This is because a smaller projection loss $\pi$ does not imply a smaller standard loss $\ell$.

**Proposition 3.6.** *Let $\ell$ be an $\ell_p$ loss, for any hypothesis $f_1, f_2$, there exists a distribution $\mathcal{D}_I$ and $\mathcal{D}$ such that $\mathbb{E}_{\mathcal{D}_I}[\pi_\ell(f_1(X), L, U)] < \mathbb{E}_{\mathcal{D}_I}[\pi_\ell(f_2(X), L, U)]$ but $\mathrm{err}(f_1) > \mathrm{err}(f_2)$.*

While minimizing the projection loss, we might overlook a hypothesis that has a smaller standard loss but a higher projection loss. However, we remark that the projection loss is still useful since it is a lower bound of the standard loss.

**Proposition 3.7.** *Let $\ell : \mathcal{Y} \times \mathcal{Y} \to \mathbb{R}$ be a loss function that satisfies Assumption 1, then for any $f$,*

$$\mathbb{E}[\pi_\ell(f(X), L, U)] \leq \mathrm{err}(f). \quad (15)$$

Consequently, if we let $\mathrm{OPT} = \mathrm{err}(f_{\mathrm{OPT}})$, we must have $f_{\mathrm{OPT}} \in \widetilde{\mathcal{F}}_{\mathrm{OPT}}$ since the projection loss is upper bound by the standard loss. This means we can apply Theorem 3.3 for $f_{\mathrm{OPT}}$ and consequently achieve an error bound similar to what we obtained in the realizable setting.

**Theorem 3.8** (Error bound, Agnostic setting). *Let $\mathcal{F}$ be a class of functions that are $m$-Lipschitz, and suppose $\ell$ satisfies Assumption 1 and the triangle inequality, then for any $f \in \widetilde{\mathcal{F}}_\eta$, we have*

$$\mathrm{err}(f) \le \mathrm{OPT} + \mathbb{E}[d(\ell, I_\eta(X), I_{\mathrm{OPT}}(X))]. \tag{16}$$

While it's not ideal to minimize the projection loss in the agnostic setting since we may not converge to $f_{\mathrm{OPT}}$, our bound suggests that the expected error of $f$ would not be much larger than that of $f_{\mathrm{OPT}}$. This error bound becomes smaller when the intervals $I_\eta(x), I_{\mathrm{OPT}}(x)$ are small. Overall, our theoretical insight suggests that we can improve our error bound by (i) having a smoother hypothesis class (smaller $m$) (ii) increasing the number of data points $n$ (which leads to smaller $\eta$), since both results in smaller intervals $I_\eta(x)$. However, if $m$ is too small, $\mathcal{F}$ may not contain a good hypothesis, causing OPT to be large.

## 4    LEARNING FROM INTERVALS USING A MINMAX OBJECTIVE

In Section 2, our goal was to learn a function $f$ that ideally lies within the given interval ($f \in \widetilde{\mathcal{F}}_0$), using an objective that penalizes values away from the given interval. In this section, we explore a different strategy: we aim to learn a function $f \in \mathcal{F}$ that minimizes the maximum loss with respect to the worst-case $\tilde{y}$ within the interval. We demonstrate that this approach yields a closed-form solution that can be evaluated efficiently. First, we define the worst-case loss as

$$\rho_\ell(f(x), l, u) := \max_{\tilde{y} \in [l, u]} \ell(f(x), \tilde{y}). \tag{17}$$

**Proposition 4.1.** *Let $\ell$ be a loss function that satisfies Assumption 1, then*

$$\rho_\ell(f(x), l, u) = 1[f(x) \le \frac{l+u}{2}]\ell(f(x), u) + 1[f(x) > \frac{l+u}{2}]\ell(f(x), l). \tag{18}$$

The proof is provided in Appendix D.5. Since $y \in [l, u]$, this objective serves as an upper bound for the true loss: $\rho_\ell(f(x), l, u) \ge \ell(f(x), y)$. Consequently, if we have a hypothesis with a small expected value $\mathbb{E}[\rho_\ell(f(x), l, u)]$, then the error $\mathrm{err}(f)$ will also be small. Based on Proposition 4.1, we define the **Minmax** objective as

$$\min_f \sum_{i=1}^n 1[f(x_i) \le \frac{l_i + u_i}{2}]\ell(f(x_i), u_i) + 1[f(x_i) > \frac{l_i + u_i}{2}]\ell(f(x_i), l_i). \tag{19}$$

In particular, when $\ell(y, y') = |y - y'|$, we can show that minimizing $\rho$ is equivalent to performing supervised learning using the mid-point of each interval.

**Corollary 4.2.** *Let $\ell(y, y') = |y - y'|$ then $\rho_\ell(f(x), l, u) = |f(x) - \frac{l+u}{2}| + \frac{u-l}{2}$ and the solution of equation 19 is equivalent to*

$$f' = \arg\min_{f \in \mathcal{F}} \sum_{i=1}^n |f(x_i) - \frac{l_i + u_i}{2}|. \tag{20}$$

The proof of this corollary is provided in Appendix D.6. This corollary establishes a connection between the heuristic of using the midpoint as a target and our approach of minimizing the maximum loss $\rho$. However, we note that $\rho$ does not take the smoothness of the hypothesis class $\mathcal{F}$ into account and may lead to the worst-case labels that are overly conservative and not reflective of the target labels. While we aim to minimize the loss with respect to the worst-case labels, we also want them to be realistic. Therefore, it would be beneficial to incorporate knowledge about certain properties of the true labels. In particular, in the realizable setting, we know that $f^* \in \widetilde{\mathcal{F}}_0$, so we may consider the worst-case labels that can be generated by some $f \in \widetilde{\mathcal{F}}_0$,

$$\min_{f \in \mathcal{F}} \max_{f' \in \widetilde{\mathcal{F}}_0} \mathbb{E}[\ell(f(X), f'(X)]. \tag{21}$$

In the realizable setting, this method also provides an upper bound for $\mathrm{err}(f)$, but it is weaker than $\rho$ because we are comparing against the worst-case $f' \in \widetilde{\mathcal{F}}_0$ rather than any possible $\tilde{y} \in [l, u]$.

**Proposition 4.3.** *In the realizable setting where $f^* \in \widetilde{\mathcal{F}}_0$, for a bounded loss $\ell$, for any $f \in \mathcal{F}$,*

$$\text{err}(f) \leq \max_{f' \in \widetilde{\mathcal{F}}_0} \mathbb{E}[\ell(f(X), f'(X))] \leq \mathbb{E}[\rho_\ell(f(X), L, U)]. \tag{22}$$

The proof is provided in Appendix D.7. With this inequality, we can conclude that when a hypothesis has a small minmax objective, its expected loss would be small as well. Moreover, we demonstrate that restricting the worst-case labels to those that could be generated by some $f \in \widetilde{\mathcal{F}}_0$ can lead to better performance than using all possible worst-case labels. This is due to worst-case labels being highly sensitive to the interval size.

**Proposition 4.4.** *For any constant $c > 0$ and $\ell(y, y') = |y - y'|$, there exists a distribution $\mathcal{D}_I$ and a hypothesis class $\mathcal{F}$ and $f^* \in \mathcal{F}$ such that for $f_1 = \arg\min_{f \in \mathcal{F}} \max_{f' \in \widetilde{\mathcal{F}}_0} \mathbb{E}[\ell(f(X), f'(X)]$ and $f_2 = \arg\min_{f \in \mathcal{F}} \mathbb{E}[\rho_\ell(f(X), L, U)], \text{err}(f_1) = 0$ while $\text{err}(f_2) > c$.*

The proof is in Appendix D.8. An empirical Minmax objective using labels from $\widetilde{\mathcal{F}}_0$ is given by

$$\min_{f \in \mathcal{F}} \max_{f' \in \widetilde{\mathcal{F}}_0} \sum_{i=1}^n \ell(f(x_i), f'(x_i)). \tag{23}$$

However, there is no closed-form solution for the inner maximization of objective in 23, making it less efficient to optimize than equation 19. To address this, we propose alternative approaches by approximately learning $f' \in \widetilde{\mathcal{F}}_0$ to solve this objective.

**1) Regularization.** We keep track of two hypothesis $f, f' \in \mathcal{F}$ and introduce a regularization term based on the projection loss to ensure that $f'$ is close $\widetilde{\mathcal{F}}_0$. We call this method **Minmax (reg)**,

$$\min_{f \in \mathcal{F}} \max_{f' \in \mathcal{F}} \sum_{i=1}^n \ell(f(x_i), f'(x_i)) - \lambda \sum_{i=1}^n \pi(f'(x_i), l_i, u_i). \tag{24}$$

Here the regularization term is always non-positive and depends only on $f'$. We can use a gradient descent ascent (Korpelevich, 1976; Chen & Rockafellar, 1997; Lin et al., 2020) algorithm that updates $f$ and $f'$ with one gradient step at a time to solve this objective.

**2) Pseudo labels.** We could replace a hypothesis class $\widetilde{\mathcal{F}}_0$ with a finite set of hypotheses $\{f_1, f_2, \ldots, f_k\}$ where $f_j \in \widetilde{\mathcal{F}}_\eta$ for some small $\eta$. We can get $f_j$ by minimizing the empirical projection loss. We then relax our objective by learning $f$ that minimizes the maximum loss with respect to $f_j$. We call this method **PL (Max)**,

$$\min_{f \in \mathcal{F}} \max_{j \in \{1, \ldots, k\}} \sum_{i=1}^n \ell(f(x_i), f_j(x_i)). \tag{25}$$

Since $f_j$ are fixed, learning $f$ becomes a minimization problem, which is more stable to solve compared to the original minmax problem. Alternatively, to further stabilize the learning objective, we can replace the max over $f_j$ with mean. We refer to this variant as **PL (Mean)**,

$$\min_{f \in \mathcal{F}} \sum_{j=1}^k \sum_{i=1}^n \ell(f(x_i), f_j(x_i)). \tag{26}$$

## 5 EXPERIMENTS

Following prior work (Cheng et al., 2023a), we conducted experiments on five public datasets from the UCI Machine Learning Repository: Abalone, Airfoil, Concrete, Housing, and Power Plant. Since these datasets are originally regression tasks with single target values, we transformed them into datasets with interval targets (described shortly). Dataset statistics are provided in Section F. For the experimental setup, we used the same configuration as (Cheng et al., 2023a): the model architecture is a MLP with hidden layers of sizes 10, 20, and 30. We trained the models using the Adam optimizer with a learning rate of 0.001 and a batch size of 512 for 1000 epochs.

**Interval Data Generation Methodology.** We propose a general approach for generating interval data for each target value $y$. This method depends on two factors: the interval size $q \in [0, \infty]$ and

| | Projection (equation 5) | Minmax (equation 19) | Minmax (reg) (equation 24) | PL (max) (equation 25) | PL (mean) (equation 26) |
|---|---|---|---|---|---|
| Abalone | $1.56_{0.01}$ | $1.65_{0.02}$ | $1.54_{0.01}$ | $\mathbf{1.52_{0.01}}$ | $\mathbf{1.52_{0.01}}$ |
| Airfoil | $\mathbf{2.46_{0.08}}$ | $2.65_{0.07}$ | $3.41_{0.04}$ | $3.31_{0.04}$ | $2.42_{0.07}$ |
| Concrete | $5.75_{0.13}$ | $7.34_{0.2}$ | $6.23_{0.16}$ | $5.86_{0.48}$ | $5.43_{0.12}$ |
| Housing | $\mathbf{5.17_{0.13}}$ | $6.88_{0.31}$ | $5.42_{0.15}$ | $5.07_{0.09}$ | $5.05_{0.09}$ |
| Power-plant | $3.4_{0.03}$ | $3.47_{0.02}$ | $3.48_{0.03}$ | $\mathbf{3.33_{0.01}}$ | $\mathbf{3.33_{0.01}}$ |
| Average (rank) | 2.8 | 4.4 | 4.2 | 2.2 | 1 |

Table 1: Test Mean Absolute Error (MAE) and the standard error (over 10 random seeds) for the uniform interval setting. PL (mean) is the best-performing method in this setting.

the interval location $p \in [0, 1]$. The interval is then defined as $[l, u] = [y - pq, y + (1 - p)q]$. When $p = 0$, the target value $y$ is at the lower boundary of the interval whereas $p = 1$ places $y$ at the upper boundary. In this work, we consider $q$ and $p$ to be generated from uniform distributions over specified ranges. The prior interval generation method in Cheng et al. (2023a) could be seen as a special case of our approach when $q \sim \text{Uniform}[0, q_{\max}]$ and $p \sim \text{Uniform}[0, 1]$.

## 5.1 RESULTS

**Which method works best in the uniform setting?** We begin by evaluating methods in the uniform interval setting described in prior work (Cheng et al., 2023a), where the interval size $q \sim \text{Uniform}[0, q_{\max}]$ and the location of the interval $p \sim \text{Uniform}[0, 1]$. For each dataset, we set $q_{\max}$ to be approximately equal to the range of the target values, $y_{\max} - y_{\min}$. Specifically, we set $q_{\max} = 30$ (Abalone), 30 (Airfoil), 90 (Concrete), 120 (Housing), and 90 (Power Plant). Our findings indicate that the PL (mean) method performs best in this uniform setting, with PL (max) and the projection method ranking second and third, respectively (Table 1). Given the superior performance of PL (mean), we conducted an ablation study to better understand its effectiveness. We explored the impact of varying the number of hypotheses $k$ and compared it with an ensemble baseline that combines pseudo-labels *before* using them to train the model, for which we still find that PL (mean) still performs better (Appendix I).

**What about other interval settings?** We conducted more detailed experiments to investigate which factors impact the performance of each method. Specifically, we varied the interval size $q$ and the interval location $p$ by 1) varying $q_{\max}$, 2) varying $q_{\min}$, 3) varying $p$ with three settings designed to position the true value $y$ at: i) *only* one boundary of the interval, ii) *both* boundaries of the interval, iii) the middle of the interval. Full details are provided in Appendix G. We found that: (1) All methods are quite robust to changes in the interval size, except for the Minmax method, whose performance decreases significantly as the interval size increases. This is consistent with our insights from the proof of 4.4), (2) The location of the true value $y$ can have a large impact on performance; specifically, the Minmax method performs better when $y$ is close to the middle of the interval. One explanation is that Minmax is equivalent to supervised learning with the midpoint of the interval (Corollary 4.2). Conversely, the other methods perform better when $y$ is close to *both* boundaries of the interval but not when $y$ is close to *only* one boundary. Finally, we conclude that if we only know that the interval size is large, it is better to use the PL (pseudo-labeling). However, if we know the true value $y$ is close to the middle of the interval, then the Minmax method is more preferable.

## 5.2 CONNECTION TO OUR THEORETICAL ANALYSIS

To validate our theoretical findings in practice, we conducted experiments designed to test whether our theory holds under empirical conditions. Recall that our main result (Theorem 3.3) states that if a hypothesis $f$ approximately lies within the intervals ($f \in \widetilde{\mathcal{F}}_\eta$) and is smooth, then $f$ will lie within intervals smaller than the original ones. To control the smoothness of our hypothesis, we utilize a Lipschitz MLP, which is an MLP augmented with spectral normalization layers (Miyato et al., 2018). The normalization ensures that the Lipschitz constant of the MLP is less than 1. We then scale the output of the MLP by a constant factor $m$ to ensure that the Lipschitz constant of the hypothesis is less than $m$.

**Reduced interval size** Our first experiment aims to determine whether the intervals, within which our hypothesis $f \in \widetilde{\mathcal{F}}_0$ lies, are smaller than the original intervals. Recall that the original in-

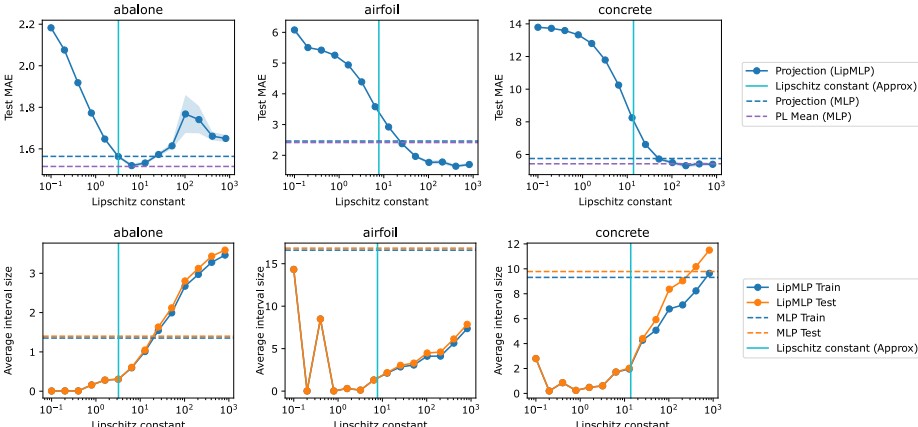

Figure 3: Test MAE of the projection method with Lipschitz MLP using different values of the Lipschitz constant. The vertical line is the Lipschitz constant approximated from the training set. (Top) The dashed horizontal lines are the test MAE of PL (Mean) and Projection approach with a standard MLP. (Bottom) Approximated interval size $I_\eta(x)$ for Lipschitz MLP with a different value of Lipschitz constant $m$. The dashed horizontal lines are the values from standard (non-Lipschitz) MLP. The figures for all datasets are in Appendix H.

tervals are given by $[l, u]$, and our theorem suggests that they would reduce to $I_\eta(x) = [\tilde{l}_{\mathcal{D} \to x}^{(m)} - r_\eta(x), \tilde{u}_{\mathcal{D} \to x}^{(m)} + s_\eta(x)]$. However, it is not possible to calculate $I_\eta(x)$ directly since it requires access to every $f \in \widetilde{\mathcal{F}}_0$. Instead, we approximate $I_\eta(x)$ using samples of hypotheses from $\widetilde{\mathcal{F}}_0$ by proceeding as follows: 1) We train 10 models with the projection objective, each from different random initializations (denoted by $f_1, \ldots, f_{10}$), 2) For each $x$, we approximate the reduced interval using the minimum and maximum values of the outputs from these models, given by $[\min_i f_i(x), \max_i f_i(x)]$. We set $m \in \{0.1, 0.1 \times 2^1, \ldots, 0.1 \times 2^{13}\}$ and consider a uniform interval setting with $q_{\max} = 90$. As expected, when the hypothesis becomes smoother, we observe that the average interval size decreases (Figure 3 (Bottom)). Moreover, we found that even when the Lipschitz constant is much larger than the value estimated from the data (vertical line), the average reduced interval size remains significantly smaller than the original interval (which is 45 since $q_{\max} = 90$). We also observe that the average interval sizes from the standard MLPs are smaller than the original values.

**Test performance** In addition to examining the average interval size, we also plot the test Mean Absolute Error (MAE) of the Lipschitz MLP with the projection objective, compared with the test MAE of the standard MLP (Figure 3 (Top)). We found that, with the right level of smoothness, Lipschitz MLP can achieve better performance than the standard MLP. When the Lipschitz constant is very small, the performance is poor for all datasets. However, performance improves as the Lipschitz constant increases. We observe that the optimal Lipschitz constant is always larger than the Lipschitz constant estimated from the training set (vertical line). For some datasets, performance degrades when the Lipschitz constant becomes too large. This aligns with our insight from Theorem 3.8, which suggests that we can improve the error bound by ensuring that the hypothesis class is as smooth as possible (smaller $m$ so that $I_\eta(x)$ is small) while still containing a good hypothesis (i.e., low OPT). Nevertheless, we do not need to know the Lipschitz constant of the dataset and can treat it as a tunable hyperparameter in practice.

## 6 CONCLUSION

In this paper, we studied the problem of learning from interval targets where only the lower and upper bounds of the target are known. We analyzed two approaches: i) learning a hypothesis that always lies within the interval, and ii) minimizing with respect to the worst-case label (or pseudolabels). Our results showed how smoothness can be beneficial by i) leading to a smaller interval, and ii) having a "regularized" worst-case label. On the experimental side, our proposed minmax/pseudolabel approach achieves good performance (where PL (mean) is the best-performing method) and validates our theoretical insights in practice. For future work, it would be interesting to study this problem for a more challenging setup with less assumption such as the true target may not always be inside the given interval or the training data are not independently and identically distributed, for instance the time-series setting.

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

## A  GENERALIZATION BOUND BASED ON THE SAMPLE SIZE $n$

The generalization bounds of Theorem 3.5 and Theorem 3.8 are non-asymptotic. The error bound is applicable for any $f \in \widetilde{\mathcal{F}}_\eta$ where the error depends on the value $\eta$. To improve the clarity of how generalization depends on the number of training sample $n$, we provide an explicit sample-complexity generalization bound for hypothesis classes whose the Rademacher complexity decay as $O(1/\sqrt{n})$. This includes a class of linear models or a class of two-layer neural networks with a bounded weight (Ma, 2022). To simplify the Theorem, we will only present the statement and the proof for the case of $L_1$ loss. However, an extension for a general $L_p$ loss is straightforward where we can replace the triangle inequality with the Minkowski's inequality.

**Theorem A.1** (Generalization bound, Realizable Setting). *Take the conditions of Theorem 3.5 (realizability and $m$-Lipschitzness) and further assume*

- *$\mathcal{F}$ is hypothesis class whose the Rademacher complexity decays as $O(1/\sqrt{n})$ e.g. two-layer neural networks with bounded weights,*

- *the support of the distribution $\mathcal{D}_I$ is a bounded set,*

- *the loss function is $\ell(y, y') = |y - y'|$.*

*Then, with probability at least $1 - \delta$, for any $f$ that minimize the empirical projection objective, for any $\tau > 0$,*

$$\mathrm{err}(f) \le \underbrace{\mathbb{E}_X[|u^{(m)}_{\mathcal{D} \to X} - l^{(m)}_{\mathcal{D} \to X}|]}_{(a)} + \tau + \underbrace{\left( \frac{D}{\sqrt{n}} + M\sqrt{\frac{\ln(1/\delta)}{n}} \right) \Gamma(\tau)}_{(b)}, \qquad (27)$$

*where $D, M$ are constants and $\Gamma(\tau) = \mathbb{E}_{\widetilde{X}} \left[ 1/\min(\mathrm{Pr}_X(lg^{(m)}_{X \to \widetilde{X}} \le \tau), \mathrm{Pr}_X(ug^{(m)}_{X \to \widetilde{X}} \le \tau)) \right]$ is a decreasing function of $\tau$.*

***Interpretation:*** Our error bound is divided into two parts.

Term (a) The first term represent an error term which depends on the smoothness property of our function class $\mathcal{F}$ and the quality of the given intervals. The first error term is **irreducible** with the number of samples in the sense that it does not decrease as we have more training samples and this error term depends solely on the quality of the intervals and the smoothness of our hypothesis class. However, this term can be small. For example, in the case when the ambiguity degree is small, this error term would be zero, ensure a perfect recovery of the true labels.

Term (b) The second and the third term capture how well we can learn a hypothesis that belongs to the intervals and these would decay as we have a larger sample size $n$. To see this, assume that we have a fix value of $\tau$, if one set $n \to \infty$ then the third term would converge to zero. That is, $(b)$ would converge to $\tau$ as $n \to \infty$. Since $\tau$ is arbitrary, we can set $\tau$ to be small so that $(b)$ would decay to zero as $n \to \infty$ and we are left with the first term $(a)$. In addition, the function $\Gamma(\tau)$ depends on the distribution of intervals $\mathcal{D}_I$. In particular, when $\mathcal{D}_I$ has small lower/upper bound gaps, $\Gamma(\tau)$ would also be small which leads to a better generalization bound for any fixed $n$.

*Proof.* Our result here can be derived from combining the results from Theorem 3.5, Proposition 3.4 and relate the Rademacher complexity of $\Pi(\mathcal{F})$ with $\mathcal{F}$. The proof here is divided into 2 steps; i) Apply the Proposition 3.4 to the Theorem 3.5 to derive the bound in terms of $\eta$, ii) Bound $\eta$ in terms of the sample size $n$.

**Step 1: Derive the bound in term of $\eta$.** Recall that from Theorem 3.5, we have

$$\text{err}(f) \leq \mathbb{E}[d(\ell, I_0(X), I_\eta(X))]. \tag{28}$$

when $I_\eta(x) = [l_{\mathcal{D}\to x}^{(m)} - r_\eta(x), u_{\mathcal{D}\to x}^{(m)} + s_\eta(x)]$. Since we have an $\ell_1$ loss, we have

$$d(\ell, I_0(x), I_\eta(x)) = |u_{\mathcal{D}\to x}^{(m)} - l_{\mathcal{D}\to x}^{(m)} + \max(r_\eta(x), s_\eta(x))|. \tag{29}$$

Substitute this back in, we have an error bound

$$\text{err}(f) \leq \mathbb{E}[|u_{\mathcal{D}\to X}^{(m)} - l_{\mathcal{D}\to X}^{(m)} + \max(r_\eta(X), s_\eta(X))|] \tag{30}$$

$$\leq \mathbb{E}[|u_{\mathcal{D}\to X}^{(m)} - l_{\mathcal{D}\to X}^{(m)}|] + \mathbb{E}[|\max(r_\eta(X), s_\eta(X))|] \quad \text{(triangle inequality).} \tag{31}$$

Now, our goal is to bound the term $\mathbb{E}[|\max(r_\eta(X), s_\eta(X))|]$. From Proposition 3.4, we know that

$$r_\eta(x) \leq \inf_\tau \tau + (\eta/\Pr(lg_{X\to x}^{(m)} \leq \tau)) \quad \text{and} \quad s_\eta(x) \leq \inf_\tau \tau + (\eta/\Pr(ug_{X\to x}^{(m)} \leq \tau)). \tag{32}$$

We place $\delta$ with $\tau$ in the original statement because we will use $\delta$ as something else, later. This implies that

$$\max(r_\eta(x), s_\eta(x)) \leq \inf_\tau \tau + \left(\frac{\eta}{\min(\Pr(lg_{X\to x}^{(m)} \leq \tau), \Pr(ug_{X\to x}^{(m)} \leq \tau))}\right). \tag{33}$$

We define $\Lambda(\mathcal{D}, \tau) = \min(\Pr(lg_{X\to x}^{(m)} \leq \tau), \Pr(ug_{X\to x}^{(m)} \leq \tau))^{-1}$ so that

$$\max(r_\eta(x), s_\eta(x)) \leq \inf_\tau \tau + \eta\Lambda(\mathcal{D}, \tau). \tag{34}$$

We can see that when $\Lambda(\mathcal{D}, \tau) \geq 0$ and $\Lambda(\mathcal{D}, \tau)$ is a decreasing function in $\tau$. Substitue this back to the equation 31, for any $\tau > 0$, we would have

$$\text{err}(f) \leq \mathbb{E}[|u_{\mathcal{D}\to X}^{(m)} - l_{\mathcal{D}\to X}^{(m)}|] + \mathbb{E}[|\tau + \eta\Lambda(\mathcal{D}, \tau)|] \tag{35}$$

$$\leq \mathbb{E}[|u_{\mathcal{D}\to X}^{(m)} - l_{\mathcal{D}\to X}^{(m)}|] + \tau + \eta\mathbb{E}[\Lambda(\mathcal{D}, \tau)] \tag{36}$$

$$= \mathbb{E}[|u_{\mathcal{D}\to X}^{(m)} - l_{\mathcal{D}\to X}^{(m)}|] + \tau + \eta\Gamma(\mathcal{D}, \tau) \tag{37}$$

where we define $\Gamma(\mathcal{D}, \tau) = \mathbb{E}[\Lambda(\mathcal{D}, \tau)]$. We can see that every term in the equation above is independent of $\eta$, apart from the term $\eta$ itself. This provide a more explicit error bound in term of $\eta$. Now, we will bound $\eta$ in terms of the number of sample $n$.

**Step 2: Bounding $\eta$ in terms of the number of sample.** Recall the result from equation 6, with probability at least $1 - \delta$ over the draws $(x_i, l_i, u_i) \sim \mathcal{D}_I$, for all $f \in \mathcal{F}$,

$$\mathbb{E}[\pi_\ell(f(X), L, U)] \leq \frac{1}{n}\sum_{i=1}^{n} \pi_\ell(f(x_i), l_i, u_i) + 2R_n(\Pi(\mathcal{F})) + M\sqrt{\frac{\ln(1/\delta)}{n}}. \tag{38}$$

Here, $R_n(\Pi(\mathcal{F}))$ is the Rademacher complexity of the function class $\Pi(\mathcal{F}) := \{\pi_\ell(f(x), l, u) \mapsto \mathbb{R} \mid f \in \mathcal{F}\}$ and we assume that the $\pi_\ell$ is uniformly bounded by $M$. We recall that we learn $\hat{f}$ by minimizing the empirical projection loss

$$\hat{f} = \arg\min_{f\in\mathcal{F}} \sum_{i=1}^{n} \pi_\ell(f(x_i), l_i, u_i). \tag{39}$$

Under the realizable setting, this objective would be zero since $f* \in \mathcal{F}$ which implies that $f*$ has zero empirical projection $\sum_{i=1}^{n} \pi_\ell(f^*(x_i), l_i, u_i) = 0$ but $\hat{f}$ also minimize the empirical projection loss so $\hat{f}$ must also have a zero empirical projection loss. We write $\eta(f)$ to refer to the $\eta$ value of $f$. Formally, defined as

$$\eta(f) = \mathbb{E}[\pi_\ell(f(X), L, U)]. \tag{40}$$

Substituting $\hat{f}$ to the bound above, we have

$$\eta(\hat{f}) \leq 2R_n(\Pi(\mathcal{F})) + M\sqrt{\frac{\ln(1/\delta)}{n}}. \tag{41}$$

The next step is to bound the Rademacher complexity $R_n(\Pi(\mathcal{F}))$ in terms of $R_n(\mathcal{F})$. We will do this by first showing that $\phi_i(f(x)) = \pi_\ell(f(x), l_i, u_i)$ is a Lipschitz continuous function and then reduce $R_n(\Pi(\mathcal{F}))$ to $R_n(\mathcal{F})$ with a variant of Talagrand's Lemma (Meir & Zhang, 2003). From our assumption that the support of $\mathcal{D}_I$ is a bounded set, and our hypothesis class is a class of two-layer neural network with bounded weight, there exists a constant $C$ for which, we have $|f(x)| \leq C$ almost surely. Here, we will show this property for $L_p$ loss, recall that

$$\phi_i(f(x)) = \pi_\ell(f(x), l_i, u_i) \tag{42}$$
$$= (l_i - f(x))^p 1[f(x) < l_i] + (f(x) - u_i)^p 1[f(x) > u]. \tag{43}$$

Differentiate with respect to $f(x)$, we have

$$|\nabla_{f(x)} \phi_i(f(x))| = p|(l_i - f(x))^{p-1} 1[f(x) < l_i] + (f(x) - u_i)^{p-1} 1[f(x) > u]| \tag{44}$$
$$\leq 2p(2C)^{p-1}. \tag{45}$$

Since this gradient is bounded for any $f(x)$, we can conclude that $\phi_i(f(x))$ is $B$-Lipschitz for some constant $B$. Now, we unpack the definition of the Rademacher complexity,

$$R_n(\Pi(\mathcal{F})) = \mathbb{E}_{(x_i, l_i, u_i) \sim \mathcal{D}_I}[\mathbb{E}_{\sigma_i \sim \{-1,1\}}[\sup_{f \in \mathcal{F}} \frac{1}{n} \sum_{i=1}^{n} \pi_\ell(f(x_i), l_i, u_i)\sigma_i]] \tag{46}$$

$$= \mathbb{E}_{(x_i, l_i, u_i) \sim \mathcal{D}_I}[\mathbb{E}_{\sigma_i \sim \{-1,1\}}[\sup_{f \in \mathcal{F}} \frac{1}{n} \sum_{i=1}^{n} \phi_i(f(x_i))\sigma_i]]. \tag{47}$$

We recall the following result from Meir & Zhang (2003) that when $\phi_1, \phi_2, \ldots \phi_n$ be functions where $\phi_i : \mathbb{R} \to \mathbb{R}$ are $\phi_i$ are $L_i$-Lipschitz, then

$$\mathbb{E}_{\sigma_i \sim \{-1,1\}}[\sup_{f \in \mathcal{F}} \frac{1}{n} \sum_{i=1}^{n} \phi_i(f(x_i))\sigma_i] \leq \mathbb{E}_{\sigma_i \sim \{-1,1\}}[\sup_{f \in \mathcal{F}} \frac{1}{n} \sum_{i=1}^{n} L_i f(x_i)\sigma_i]. \tag{48}$$

Applying this result with the fact that $\phi_i$ is $B$-Lipschitz for all $i = 1, \ldots, n$, we can conclude that

$$R_n(\Pi(\mathcal{F})) = \mathbb{E}_{(x_i, l_i, u_i) \sim \mathcal{D}_I}[\mathbb{E}_{\sigma_i \sim \{-1,1\}}[\sup_{f \in \mathcal{F}} \frac{1}{n} \sum_{i=1}^{n} \phi_i(f(x_i))\sigma_i]] \tag{49}$$

$$\leq \mathbb{E}_{(x_i, l_i, u_i) \sim \mathcal{D}_I}[\mathbb{E}_{\sigma_i \sim \{-1,1\}}[\sup_{f \in \mathcal{F}} \frac{1}{n} \sum_{i=1}^{n} Bf(x_i)\sigma_i]] \tag{50}$$

$$= BR_n(\mathcal{F}). \tag{51}$$

We successfully reduce the Rademacher complexity of $\Pi(\mathcal{F})$ to $\mathcal{F}$. Since we assume that the Rademacher complexity of $\mathcal{F}$ decays as $O(1/\sqrt{n})$, there exists a constant $D$ such that

$$R_n(\Pi(\mathcal{F})) \leq \frac{D}{\sqrt{n}} \tag{52}$$

and

$$\eta(\hat{f}) \leq \frac{D}{\sqrt{n}} + M\sqrt{\frac{\ln(1/\delta)}{n}} \tag{53}$$

for some constant $D, M$. Substitute this back to the result from step 1 concludes our proof. $\square$

In the general setting where we have $\ell(y, y') = |y - y'|^p$, we would have the following error bound

$$\mathrm{err}(f) \leq \left( \mathbb{E}_X[|u_{\mathcal{D} \to X}^{(m)} - l_{\mathcal{D} \to X}^{(m)}|^p]^{1/p} + \tau + \left( \frac{D}{\sqrt{n}} + M\sqrt{\frac{\ln(1/\delta)}{n}} \right)^{1/p} \Gamma(\tau)^{1/p} \right)^p \tag{54}$$

We can also write this generalization bound for the agnostic setting

**Theorem A.2** (Generalization Bound, Agnostic Setting). *Under the conditions of Theorem A.1 apart from realizability, with probability at least $1 - \delta$, for any $f$ that minimize the empirical projection objective, for any $\tau > 0$,*

$$\text{err}(f) \leq \underbrace{\text{OPT}}_{(a)} + \underbrace{\mathbb{E}_X[|u_{\mathcal{D}\to X}^{(m)} - l_{\mathcal{D}\to X}^{(m)}|]}_{(b)} + \tau + \underbrace{\left(\widehat{\text{err}}(f) + \frac{D}{\sqrt{n}} + M\sqrt{\frac{\ln(1/\delta)}{n}} + \text{OPT}\right)\Gamma(\tau)}_{(c)},$$

(55)

*where $D, M$ are constants and $\Gamma(\tau) = \mathbb{E}_{\widetilde{X}}\left[1/\min(\text{Pr}_X(lg_{X\to\widetilde{X}}^{(m)} \leq \tau), \text{Pr}_X(ug_{X\to\widetilde{X}}^{(m)} \leq \tau))\right]$ is a decreasing function of $\tau$, $\widehat{\text{err}}(f)$ is an empirical projection error of $f$, and $\text{OPT}$ is the expected error of the optimal hypothesis in $\mathcal{F}$.*

***Interpretation:*** In contrast to the realizability setting, our error bound for the agnostic setting is divided into three parts.

Term (a) The first term represent an error term of the optimal hypothesis in $\mathcal{F}$, given by $\text{OPT}$.

Term (b) The second term represent an error term which depends on the smoothness property of our function class $\mathcal{F}$ and the quality of the given intervals similar to the realizability setting.

Term (c) The third and the fourth term capture how well we can learn a hypothesis that belongs to the intervals. The key difference between this agnostic setting and the realizability setting is that this term would not decay to zero anymore as $n \to \infty$. In particular, for a fixed $\tau$, we can see that as $n \to \infty$, we would have $\widehat{\text{err}}(f) \to \text{OPT}$ since we are minimizing the empirical projection loss and as a result, this third part would converge to

$$\tau + 2\,\text{OPT}\,\Gamma(\tau). \tag{56}$$

Since this hold for any $\tau$, the optimal $\tau$ would be the one such that $\tau = 2\,\text{OPT}\,\Gamma(\tau)$ and this value depends on the distribution $\mathcal{D}_I$.

Overall, when $n \to \infty$, the upper bound would converge to

$$\text{OPT} + \mathbb{E}_X[|u_{\mathcal{D}\to X}^{(m)} - l_{\mathcal{D}\to X}^{(m)}|] + \tau + 2\,\text{OPT}\,\Gamma(\tau). \tag{57}$$

This can be small as long as the $\text{OPT}$ is small, the expected lower/ upper bound gaps are small and when the noise in the given intervals are small. The proof of this theorem follows the same argument from the realizable setting.

# B RELAXATION OF AMBIGUITY DEGREE FOR A REGRESSION SETTING

As noted in the related work section, the ambiguity degree is defined in the context of classification and it might not be suitable for regression tasks. This is due to the nature of the loss function, In classification, a hypothesis is either correct or incorrect, and a small ambiguity degree ensures that we can recover the true label. However, in regression, we are often satisfied with predictions that are sufficiently close to the target—for example, within an error tolerance of $\epsilon$. This implies that we do not need to recover the exact true label, but a ball with a small radius around the true label might be sufficient.

In this section, we explore a relaxation of the original ambiguity degree to the regression setting. Motivated by the concept of a tolerable area around the true label $y$, we define an ambiguity radius

**Definition B.1** (Ambiguity Radius). *For distributions $\mathcal{D}, \mathcal{D}_I$ with a probability density function $p$, an ambiguity radius is defined as*

$$\text{AmbiguityRadius}(\mathcal{D}, \mathcal{D}_I) := \min_{r \geq 0} r \quad s.t. \quad \Pr_{X,Y\sim\mathcal{D}}\left(\bigcap_{p(X,l,u)>0}[l,u] \subseteq B(Y,r)\right) = 1 \tag{58}$$

*when $B(y,r) = \{y' \mid |y - y'| \leq r\}$ is a ball of radius $r$ around $y$.*

The interpretation of this is that it is the smallest radius $r$ for which we are guaranteed the intersection of all interval for a given $x$ must lie within a radius of $r$ from the true label $y$. As a direct consequence, we know that whenever the ambiguity degree is small the ambiguity radius must be zero since the intersection of all interval for a given $x$ is just the true label $\{y\}$.

In fact, our analysis have captured the essence of this interval intersection for each $x$. We recall that for any $f \in \widetilde{\mathcal{F}}_0$ and for each $x$ with $p(x) > 0$,

$$f(x) \in I_0(x) = [l^{(m)}_{\mathcal{D} \to x}, u^{(m)}_{\mathcal{D} \to x}] \subseteq B(y, r^*), \tag{59}$$

when $r^*$ is the ambiguity radius. This follows directly from the definition of the ambiguity radius. As a result, we know that each interval $I_0(x)$ would have a size at most $2r^*$. The same technique as in the Section 3.3 would imply that the expected error of any $f \in \widetilde{\mathcal{F}}_0$ would be at most $2r^*$ in the realizable setting (with $L_1$ loss).

Finally, we want to remark that our analysis not only is applicable to this extension of the ambiguity degree to the ambiguity radius, we further use the smooth property of $\mathcal{F}$ and $I_0(x)$ might even be a proper subset of the ball $B(y, r^*)$, giving a result stronger than one based solely on the ambiguity radius.

## C  RELATED WORK

**Weak supervision.** Our setting is part of a sub-field of weak supervision where one learns from noisy, limited, or imprecise sources of data rather than a large amount of labeled data. Learning from noisy labels assumes that we only observe a noisy version of the true labels at the training time where the noise follows different noise models (usually random noise) (Natarajan et al., 2013; Li et al., 2017; Song et al., 2022; Angluin & Laird, 1988; Karimi et al., 2020; Awasthi et al., 2017; Chen et al., 2019; Long & Servedio, 2008; Diakonikolas et al., 2019). Programmatic weak supervision, on the other hand, assumes that we have access to multiple noisy weak labels (but deterministic noise) specified by domain experts, e.g. from logic rules or heuristics methods (Zhang et al., 2022; Zhang et al.; Ratner et al., 2016; 2017; Rühling Cachay et al., 2021; Shin et al., 2022; Karamanolakis et al., 2021; Fu et al., 2020; Pukdee et al., 2023b). Positive-unlabeled learning is another type of weak supervision where the training set only contains positive examples and unlabeled examples (Kiryo et al., 2017; Du Plessis et al., 2014; Bekker & Davis, 2020; Elkan & Noto, 2008; Li & Liu, 2003; Hsieh et al., 2015).

**Learning with side information.** In contrast to the weakly supervised setting, we have access to standard labeled data but also have access to some additional information. This could be unlabeled data which is studied in semi-supervised learning (Zhu, 2005; Chapelle et al.; Kingma et al., 2014; Van Engelen & Hoos, 2020; Berthelot et al., 2019; Zhu & Goldberg, 2022; Laine & Aila, 2016; Zhai et al., 2019; Sohn et al., 2020; Yang et al., 2016) or different constraints based on the domain knowledge such as physics rules (Willard et al., 2020; Swischuk et al., 2019; Karniadakis et al., 2021; Wu et al., 2018; Kashinath et al., 2021) or explanations (Ross et al., 2017; Pukdee et al., 2023a; Rieger et al., 2020; Erion et al., 2021) or output constraints (Yang et al., 2020; Brosowsky et al., 2021) which is similar to the interval targets. In some settings, interval targets are the best thing one could have (similar to the weak supervision setting) but in many cases such as in bond pricing, target intervals are readily available in the wild and could also be considered as a side information.

## D  ADDITIONAL PROOFS

### D.1  PROOF OF PROPOSITION 2.1

*Proof.* First, we assume that $\pi_\ell(f(x), l, u) = 0$. This implies that there exists $\tilde{y} \in [l, u]$ such that $\ell(f(x), \tilde{y}) = 0$. From the assumption on $\ell$ that $\ell(y, y') = 0$ if and only if $y = y'$, we must have $f(x) = \tilde{y} \in [l, u]$ as required. On the other hand, if $f(x) \in [l, u]$, it is clear that $\pi_\ell(f(x), l, u) = \ell(f(x), f(x)) = 0$ since $\ell(y, y') \geq 0$.

Now, assume that we can write $\ell(y, y') = \psi(|y - y'|)$ for some non-decreasing function $\psi$, we have

$$\pi_\ell(f(x), l, u) = \min_{\tilde{y} \in [l,u]} \psi(|f(x) - \tilde{y}|) \tag{60}$$

$$= \psi(\min_{\tilde{y} \in [l,u]} |f(x) - \tilde{y}|) \tag{61}$$

$$= \begin{cases} \psi(l - f(x)) & f(x) < l \\ \psi(0) & l \le f(x) \le u \\ \psi(f(x) - u) & f(x) > u \end{cases} \tag{62}$$

$$= 1[f(x) < l]\ell(f(x), l) + 1[f(x) > u]\ell(f(x), u). \tag{63}$$

Here we rely on the assumption that $\psi$ is non-decreasing so the minimum value of $\psi(x)$ happens when $x$ is also at the minimum value. $\qquad\square$

### D.2 Proof of Proposition 3.6

*Proof.* Since $f_1 \ne f_2$, there exists $x$ such that $f_1(x) \ne f_2(x)$. Without loss of generality, let $f_1(x) < f_2(x)$. Consider a simple one point distribution $\mathcal{D}$ with only one data point $(x, y) = (x, f_2(x) + \epsilon)$ with probability mass 1 and $\mathcal{D}_I$ be another one point distribution with $(x, l, u) = (x, f(x_1) - \epsilon, f(x_2) - \epsilon)$. We can see that $0 = \mathbb{E}_{\mathcal{D}_I}[\pi(f_1(X), L, U)] < \mathbb{E}_{\mathcal{D}_I}[\pi(f_2(X), L, U)] = \epsilon^p$ while $(f(x_2) - f(x_1) + \epsilon)^p = \text{err}(f_1) > \text{err}(f_2) = \epsilon^p$. $\qquad\square$

### D.3 Proof of Proposition 3.7

*Proof.* From the Proposition 2.1,

$$\pi(f(x), l, u) = 1[f(x) < l]\ell(f(x), l) + 1[f(x) > u]\ell(f(x), u) \tag{64}$$

Recall that $y \in [l, u]$, we consider 3 cases,

1. $f(x) < l$, $\pi(f(x), l, u) = \ell(f(x), l) = \psi(|l - f(x)|) \le \psi(|y - f(x)|) = \ell(f(x), y)$

2. $f(x) > u$, $\pi(f(x), l, u) = \ell(f(x), u) = \psi(|f(x) - u|) \le \psi(|f(x) - y|) = \ell(f(x), y)$

3. $l \le f(x) \le u$, $\pi(f(x), l, u) = 0 \le \ell(f(x), y)$

$\qquad\square$

### D.4 Proof of Theorem 3.8

*Proof.* From the triangle inequality,

$$\ell(f(x), y) = \ell(f(x), f_{\text{OPT}}(x)) + \ell(f_{\text{OPT}}(x), y) \tag{65}$$

We can take an expectation to have

$$\mathbb{E}[\ell(f(X), Y)] \le \mathbb{E}[\ell(f(X), f_{\text{OPT}}(X)] + \text{OPT}. \tag{66}$$

Since $f_{\text{OPT}} \in \widetilde{\mathcal{F}}_{\text{OPT}}$ which from Theorem 3.3, we can bound

$$f_{\text{OPT}}(x) \in [l_{\mathcal{D} \to x}^{(m)} - r_{\text{OPT}}(x), l_{\mathcal{D} \to x}^{(m)} + s_{\text{OPT}}(x)]. \tag{67}$$

Similarly, for any $f \in \widetilde{\mathcal{F}}_\eta$, we have

$$f(x) \in [l_{\mathcal{D} \to x}^{(m)} - r_\eta(x), u_{\mathcal{D} \to x}^{(m)} + s_\eta(x)] \tag{68}$$

Finally, we can bound the error between any two intervals with the maximum loss between their boundaries. $\qquad\square$

## D.5 Proof of Proposition 4.1

*Proof.* Since we can write $\ell(y, y') = \psi(|y - y'|)$ for some non-decreasing function $\psi$, we have

$$\rho_\ell(f(x), l, u) = \max_{\tilde{y} \in [l,u]} \psi(|f(x) - \tilde{y}|) \tag{69}$$

$$= \psi(\max_{\tilde{y} \in [l,u]} |f(x) - \tilde{y}|) \tag{70}$$

$$= \begin{cases} \psi(u - f(x)) & f(x) < \frac{l+u}{2} \\ \psi(f(x) - l) & f(x) \geq \frac{l+u}{2} \end{cases} \tag{71}$$

$$= 1[f(x) \leq \frac{l+u}{2}]\ell(f(x), u) + 1[f(x) > \frac{l+u}{2}]\ell(f(x), l). \tag{72}$$

Here we rely on the assumption that $\psi$ is non-decreasing so the maximum value of $\psi(x)$ happens when $x$ is also at the maximum value. $\qquad \square$

## D.6 Proof of Corollary 4.2

*Proof.* Since $\ell(y, y') = |y - y'|$, from Proposition 4.1, we have a closed form solution of $\rho$,

$$\rho_\ell(f(x), l, u) = 1[f(x) \leq \frac{l+u}{2}]\ell(f(x), u) + 1[f(x) > \frac{l+u}{2}]\ell(f(x), l) \tag{73}$$

$$= 1[f(x) \leq \frac{l+u}{2}](u - f(x)) + 1[f(x) > \frac{l+u}{2}](f(x) - l) \tag{74}$$

$$= 1[f(x) \leq \frac{l+u}{2}](u - \frac{l+u}{2} + \frac{l+u}{2} - f(x)) + 1[f(x) > \frac{l+u}{2}](f(x) - \frac{l+u}{2} + \frac{l+u}{2} - l) \tag{75}$$

$$= \frac{u-l}{2} + 1[f(x) \leq \frac{l+u}{2}](\frac{l+u}{2} - f(x)) + 1[f(x) > \frac{l+u}{2}](f(x) - \frac{l+u}{2}) \tag{76}$$

$$= |f(x) - \frac{l+u}{2}| + \frac{u-l}{2}. \tag{77}$$

Since $u_i, l_i$ are constants, $\frac{u_i - l_i}{2}$ would have no impact on the optimal solution of equation 19 and therefore, the optimal would also be the same as the one that minimizes $\sum_{i=1}^{n} |f(x_i) - \frac{l_i + u_i}{2}|$. $\quad \square$

## D.7 Proof of Proposition 4.3

*Proof.* From the realizability assumption, we know that $f^* \in \widetilde{\mathcal{F}}_0$, therefore,

$$\text{err}(f) = \mathbb{E}[\ell(f(X), f^*(X))] \leq \max_{f' \in \widetilde{\mathcal{F}}_0} \mathbb{E}[\ell(f(X), f'(X))]. \tag{78}$$

On the other hand, Let $f'' \in \widetilde{\mathcal{F}}_0$, be a hypothesis that achieves the maximum value of $\mathbb{E}[\ell(f(X), f''(X))]$. Since $f'' \in \widetilde{\mathcal{F}}_0$ we know that

$$\mathbb{E}[\pi_\ell(f''(X), L, U)] = 0. \tag{79}$$

Since the projection loss is always non-negative and is continuous, from Lemma E.1, we can conclude that $\pi_\ell(f''(x), l, u) = 0$ for any $x, l, u$ with positive density function $p(x, l, u) > 0$ which implies $f''(x) \in [l, u]$. Therefore, for any $x$ with $p(x) > 0$,

$$\ell(f(x), f''(x)) \leq \max_{\tilde{y} \in [l,u]} \ell(f(x), \tilde{y}) = \rho_\ell(f(x), l, u). \tag{80}$$

We can take an expectation over $X, L, U$ and have the desired result. $\qquad \square$

## D.8 Proof of Proposition 4.4

*Proof.* Consider when $\mathcal{X} = \{0, 1\}$ and $f^*$ such that $f^*(0) = f^*(1) = 0$. Consider a hypothesis class of constant functions $\mathcal{F} = \{f : \mathcal{X} \to \mathbb{R} \mid f(x) = d, \forall x \in \mathcal{X}\}$. We can see that $f^* \in \mathcal{F}$. Assume that we have a uniform distribution over $\mathcal{X}$ and we also have deterministic interval

$[l(x), u(x)]$. Assume that for $x = 0$, we have an interval $[l(0), u(0)] = [-a, \epsilon]$ for some $a > 0$ and for $x = 1$, we have an interval $[l(1), u(1)] = [-\epsilon, 2\epsilon]$. Since $\mathcal{F}$ is a class of constant hypothesis, for all $x$, we must have $f(x) \in [-a, \epsilon] \cap [-\epsilon, 2\epsilon] = [-\epsilon, \epsilon]$. This implies that

$$\widetilde{\mathcal{F}}_0 = \{f \mid f(x) = c, \forall x \in \mathcal{X}, c \in [-\epsilon, \epsilon]\}. \tag{81}$$

Therefore,

$$f_1 = \arg \min_{f \in \mathcal{F}} \max_{f' \in \widetilde{\mathcal{F}}_0} \mathbb{E}[\ell(f(X), f'(X)] \tag{82}$$

$$= \arg \min_{f \in \mathcal{F}} \max_{f' \in \widetilde{\mathcal{F}}_0} \frac{1}{2}(|f(0) - f'(0)| + |f(1) - f'(1)|) \tag{83}$$

$$= \arg \min_{f \in \mathcal{F}} \max_{c \in [-\epsilon, \epsilon]} |f(0) - c| \tag{84}$$

$$\tag{85}$$

By symmetry, we can see that the optimal $f_1(x) = 0$ which means that $\text{err}(f_1) = 0$. On the other hand, consider $f_2$, from Corollary 4.2, $f_2$ is equivalent to the solution of supervised learning with the midpoint of each interval,

$$f_2 = \arg \min_{f \in \mathcal{F}} \mathbb{E}[\rho_\ell(f(X), L, U)] \tag{86}$$

$$= \arg \min_{f \in \mathcal{F}} \frac{1}{2}[|f(0) - \frac{-a + \epsilon}{2}| + |f(1) - \frac{-\epsilon + 2\epsilon}{2}|]. \tag{87}$$

By symmetry, the optimal $f_2$ should lie in the middle between these two points so that $f_2(x) = -a/2 + \epsilon$. We would have $\text{err}(f_2) = |-a/2 + \epsilon|$ which can be arbitrarily large as $a \to \infty$. □

# E  PROBABILISTIC INTERVAL SETTING

In this section, we consider the probabilistic interval setting which is when, for each $x$, the corresponding interval is drawn from some distribution $\mathcal{D}_I$. We assume that $\mathcal{D}_I$ is a nonatomic distribution i.e. it does not contain a point mass. We also use $p$ to refer to the probability density function.

**Assumption 2.** *A distribution $P$ with a probability density function $p(x)$ is a nonatomic distribution when for any $x$ such that $p(x) > 0$ and for any $\epsilon > 0$, there exists a set $S_{x,\epsilon} \subseteq B(x, \epsilon)$ (a ball with radius $\epsilon$) such that $\Pr(S_{x,\epsilon}) > 0$. We assume that the distribution $\mathcal{D}$ and $\mathcal{D}_I$ are nonatomic distributions.*

**Lemma E.1.** *Let $P$ be a nonatomic distribution over $\mathcal{X}$ with a probability density function $p(x)$. For any continuous function $f : \mathcal{X} \to [0, \infty)$, if $\mathbb{E}_P[f(X)] = 0$ then $f(x) = 0$ for all $x$ with $p(x) > 0$.*

*Proof.* We will prove this by contradiction. Assume that there exists $x$ with $p(x) > 0$ such that $f(x) > 0$. By the continuity of $f$, there exists $\delta_1 > 0$ such that for any $x' \in B(x, \delta_1)$ such that $|f(x) - f(x')| \leq f(x)/2$ which implies that $f(x') \geq f(x)/2$. In addition, by the nonatomic assumption, there exists $S_{x,\delta_1} \subseteq B(x, \delta_1)$ such that $\Pr(S_{x,\delta_1}) > 0$. Therefore,

$$\mathbb{E}_P[f(X)] = \int_{w \in \mathcal{X}} f(w)p(w)dw \tag{88}$$

$$\geq \int_{w \in S_{x,\delta_1}} f(w)p(w)dw \tag{89}$$

$$\geq \int_{w \in S_{x,\delta_1}} \frac{f(x)p(w)}{2}dw \tag{90}$$

$$= \frac{f(x)\Pr(S_{x,\delta_1})}{2} > 0. \tag{91}$$

This leads to a contradiction since $\mathbb{E}_P[f(X)] > 0$. □

Similar to the deterministic interval setting, for any $f \in \widetilde{\mathcal{F}}_0$, $f$ has to lie inside the interval as well. One difference would be that in the probabilistic interval setting, we can have multiple intervals for each $x$ and since $f$ has to lie inside all of them, $f$ would also lie inside the intersection of all of them for which we denote as $[\tilde{l}_x, \tilde{u}_x]$ for each $x$.

**Proposition E.2.** *For any $f \in \widetilde{\mathcal{F}}_0$, and a loss function $\ell$ that satisfies Assumption 1, for any $x$ with positive probability density $p(x) > 0$, we have*

$$f(x) \in \bigcap_{p(x,l,u)>0} [l, u] := [\tilde{l}_x, \tilde{u}_x]. \tag{92}$$

*Proof.* Let $f \in \widetilde{\mathcal{F}}_0$ so we have $\mathbb{E}[\pi(f(X), L, U)] = 0$. From Lemma E.1, for any $(x, l, u)$ such that $p(x, l, u) > 0$, we have $\pi(f(x), l, u) = 0$ which implies $f(x) \in [l, u]$ (From Proposition 2.1). Therefore, by taking an intersection over all possible intervals, we would have $f(x) \in \bigcap_{p(x,l,u)>0}[l, u] := [\tilde{l}_x, \tilde{u}_x]$. □

**Proposition E.3.** *Let $\mathcal{F}$ be a class of functions that are $m$-Lipschitz. For any $x, x'$, denote $\tilde{l}_{x' \to x}^{(m)} = \tilde{l}_{x'} - m\|x - x'\|$, $\tilde{u}_{x' \to x}^{(m)} = \tilde{u}_{x'} + m\|x - x'\|$, then for any $f \in \widetilde{\mathcal{F}}_0$ and for any $x$ with positive probability density $p(x) > 0$,*

$$f(x) \in \bigcap_{x'}[\tilde{l}_{x' \to x}^{(m)}, \tilde{u}_{x' \to x}^{(m)}] := [\tilde{l}_{\mathcal{D} \to x}^{(m)}, \tilde{u}_{\mathcal{D} \to x}^{(m)}] \tag{93}$$

*Proof.* Consider $f \in \widetilde{\mathcal{F}}_0$, since $f$ is $m$-Lipschitz, for any $x, x' \in \mathcal{X}$, we have $|f(x) - f(x')| \leq m\|x - x'\|$ which implies

$$f(x') - m\|x - x'\| \leq f(x) \leq f(x') + m\|x - x'\| \tag{94}$$

We illustrate this in Figure 2a. Then, from Proposition E.2, for $f \in \widetilde{\mathcal{F}}_0$, we have $\tilde{l}_{x'} \leq f(x') \leq \tilde{u}_{x'}$ which implies

$$\tilde{l}_{x' \to x}^{(m)} = \tilde{l}_{x'} - m\|x - x'\| \leq f(x') - m\|x - x'\| \tag{95}$$

$$\tilde{u}_{x' \to x}^{(m)} = \tilde{u}_{x'} + m\|x - x'\| \geq f(x') - m\|x + x'\|. \tag{96}$$

Substitute back to equation equation 94 and take supremum over $x'$, we have

$$\tilde{l}_{x' \to x}^{(m)} \leq f(x) \leq \tilde{u}_{x' \to x}^{(m)} \tag{97}$$

$$\sup_{x'} \tilde{l}_{x' \to x}^{(m)} \leq f(x) \leq \inf_{x'} \tilde{u}_{x' \to x}^{(m)} \tag{98}$$

$$\tilde{l}_{\mathcal{D} \to x}^{(m)} \leq f(x) \leq \tilde{u}_{\mathcal{D} \to x}^{(m)}. \tag{99}$$

□

Next, we present the probabilistic interval version of Theorem 3.3. Details of the proofs are the same, except that we use $\tilde{l}, \tilde{u}$ instead of $l, u$.

**Theorem E.4.** *Let $\mathcal{F}$ be a class of functions that are $m$-Lipschitz. $\ell : \mathcal{Y} \times \mathcal{Y} \to \mathbb{R}$ is a loss function that satisfies Assumption 1. For any $f \in \widetilde{\mathcal{F}}_\eta$ and for any $x$ with positive probability density $p(x) > 0$,*

$$f(x) \in [\tilde{l}_{\mathcal{D} \to x}^{(m)} - r_\eta(x), \tilde{u}_{\mathcal{D} \to x}^{(m)} + s_\eta(x)] \tag{100}$$

*where $\tilde{l}_{\mathcal{D} \to x}^{(m)}, \tilde{u}_{\mathcal{D} \to x}^{(m)}$ are defined as in Proposition E.3 and*

1. *$r_\eta(x) = r$ such that $\eta = \mathbb{E}[1[g(x, X, r) < L]\ell(g(x, X, r), L)]$ where $g(x, x', r) = \tilde{l}_{x'} - (r - (\tilde{l}_{\mathcal{D} \to x}^{(m)} - \tilde{l}_{x' \to x}^{(m)}))$.*

2. *$s_\eta(x) = s$ such that $\eta = \mathbb{E}[1[h(x, X, s) > U]\ell(h(x, X, s), U)]$ where $h(x, x', s) = \tilde{u}_{x'} + (s - (\tilde{u}_{x' \to x}^{(m)} - \tilde{u}_{\mathcal{D} \to x}^{(m)}))$.*

*Proof.* Now, we will show that if $f \in \widetilde{\mathcal{F}}_\eta$ then we have $f(x) \in [\tilde{l}_{\mathcal{D}\to x}^{(m)} - r_\eta(x), \tilde{u}_{\mathcal{D}\to x}^{(m)} + s_\eta(x)]$ instead. First, we explore what would be a requirement to change the lower bound of $f(x)$ from $\tilde{l}_{\mathcal{D}\to x}^{(m)}$ to $\tilde{l}_{\mathcal{D}\to x}^{(m)} - r$. Again, from Lipschitzness,

$$f(x') - m\|x - x'\| \leq f(x) \tag{101}$$

Taking a supremum here, we have

$$\sup_{x'} f(x') - m\|x - x'\| \leq f(x). \tag{102}$$

Here, we will use $\sup_{x'} f(x') - m\|x - x'\|$ as a new lower bound for $f(x)$. Assume that it is lower than $\tilde{l}_{\mathcal{D}\to x}^{(m)}$, we can write

$$\sup_{x'} f(x') - m\|x - x'\| = \tilde{l}_{\mathcal{D}\to x}^{(m)} - r \tag{103}$$

for some $r > 0$, then it implies that for all $x' \in \mathcal{X}$, we must have

$$f(x') - m\|x - x'\| \leq \tilde{l}_{\mathcal{D}\to x}^{(m)} - r \tag{104}$$

$$(f(x') - \tilde{l}_{x'} + (\tilde{l}_{x'} - m\|x - x'\|) \leq \tilde{l}_{\mathcal{D}\to x}^{(m)} - r \tag{105}$$

$$f(x') \leq \tilde{l}_{x'} - \tilde{l}_{x'\to x}^{(m)} + \tilde{l}_{\mathcal{D}\to x}^{(m)} - r \tag{106}$$

$$f(x') \leq \tilde{l}_{x'} - (r - (\tilde{l}_{\mathcal{D}\to x}^{(m)} - \tilde{l}_{x'\to x}^{(m)})) \tag{107}$$

That is, if one can change the lower bound of $f(x)$ from $\tilde{l}_{\mathcal{D}\to x}^{(m)}$ to $\tilde{l}_{\mathcal{D}\to x}^{(m)} - r$ then for all $x'$, $f(x')$ has to take value lower than $\tilde{l}_{x'}$ by at least $r - (\tilde{l}_{\mathcal{D}\to x}^{(m)} - \tilde{l}_{x'\to x}^{(m)})$ whenever this term is positive. However, $f \in \widetilde{\mathcal{F}}_\eta$ so that $f(x')$ can't be too far away from $\tilde{l}_{x'}$ since $\mathbb{E}[\pi_\ell(f(X), L, U)] \leq \eta$. From Proposition 2.1, if one can write $\ell(y, y') = \psi(|y - y'|)$ for some non-decreasing function $\psi$ then we have

$$\pi_\ell(f(x), l, u) = 1[f(x) < l]\ell(f(x), l) + 1[f(x) > u]\ell(f(x), u). \tag{108}$$

Therefore,

$$\eta \geq \mathbb{E}[\pi_\ell(f(X), L, U)] \geq \mathbb{E}[1[f(X) < L]\ell(f(X), L)]. \tag{109}$$

Let $g(x, x', r) = \tilde{l}_{x'} - (r - (\tilde{l}_{\mathcal{D}\to x}^{(m)} - \tilde{l}_{x'\to x}^{(m)}))$ be the upper bound of $f(x')$ for any $x'$ as we derived in the equation equation 107. Since $1[a < L]\ell(a, L)$ is a decreasing function over $a$, equation 109 implies

$$\eta \geq \mathbb{E}[1[f(X) < L]\ell(f(X), L)] \geq \mathbb{E}[1[g(x, X, r) < L]\ell(g(x, X, r), L)] \tag{110}$$

We can also see that $g(x, x', r)$ is a decreasing function of $r$ which means $\mathbb{E}[1[g(x, X, r) < L]\ell(g(x, X, r), L)]$ is an increasing function of $r$. The largest possible value of $r$ would then be the $r$ such that the inequality holds,

$$\eta = \mathbb{E}[1[g(x, X, r) < L]\ell(g(x, X, r), L)]. \tag{111}$$

which we denoted this as $r_\eta(x)$. Similarly, we can show that if the largest possible value of $s$ such that we can change the upper bound of $f(x)$ from $\tilde{u}_{\mathcal{D}\to x}^{(m)}$ to $\tilde{u}_{\mathcal{D}\to x}^{(m)} + s$ is given by

$$\eta = \mathbb{E}[1[h(x, X, s) > U]\ell(h(x, X, s), U)] \tag{112}$$

where $h(x, x', s) = \tilde{u}_{x'} + (s - (\tilde{u}_{x'\to x}^{(m)} - \tilde{u}_{\mathcal{D}\to x}^{(m)}))$. $\qquad\square$

**Theorem E.5.** *Under the conditions of Theorem E.4, if further assume that for each $x$, the lower and upper bound of $y$ is given by deterministic function $[l(x), u(x)]$ and $\ell$ is an $\ell_p$ loss $\ell(y, y') = |y - y'|^p$ and denote the lower bound gap and upper bound gap of $f(x)$ induced by $x'$ as $lg_{x'\to x}^{(m)} = \tilde{l}_{\mathcal{D}\to x}^{(m)} - \tilde{l}_{x'\to x}^{(m)}$ and $ug_{x'\to x}^{(m)} = \tilde{u}_{x'\to x}^{(m)} - \tilde{u}_{\mathcal{D}\to x}^{(m)}$ then we have*

$$r_\eta(x) = r \quad s.t. \quad \mathbb{E}[(r - lg_{X\to x}^{(m)})_+^p] = \eta \tag{113}$$

$$s_\eta(x) = s \quad s.t. \quad \mathbb{E}[(s - ug_{X\to x}^{(m)})_+^p] = \eta \tag{114}$$

*where we denote $c_+ = \max(0, c)$. Further, we can bound $r_\eta(x)$ and $s_\eta(x)$,*

$$r_\eta(x) \leq \inf_\delta \delta + \left(\frac{\eta}{\Pr(lg_{X\to x}^{(m)} \leq \delta)}\right)^{1/p} \tag{115}$$

$$s_\eta(x) \leq \inf_\delta \delta + \left(\frac{\eta}{\Pr(ug_{X\to x}^{(m)} \leq \delta)}\right)^{1/p}. \tag{116}$$

*Proof.* Since $[l, u]$ is deterministic for each $x$, we have $\tilde{l}_x = l(x)$. By the property of squared loss,

$$\mathbb{E}[1[g(x, X, r) < L]\ell(g(x, X, r), L)] = \mathbb{E}[(L - g(x, X, r))_+^p] \tag{117}$$

$$= \mathbb{E}[(l(X) - g(x, X, r))_+^p] \tag{118}$$

$$= \mathbb{E}[(l(X) - (\tilde{l}_X - (r - (\tilde{l}_{\mathcal{D} \to x}^{(m)} - \tilde{l}_{X \to x}^{(m)}))))_+^p] \tag{119}$$

$$= \mathbb{E}[(r - lg_{X \to x}^{(m)})_+^p] \tag{120}$$

as required. We can use a similar argument for $s_\eta(x)$. Next, we can see that for any valid value of $r$,

$$\eta \geq \mathbb{E}[(r - lg_{X \to x}^{(m)})_+^p] \geq \mathbb{E}[(r - \delta)_+^p 1[lg_{X \to x}^{(m)} \leq \delta]] = (r - \delta)_+^p \Pr(lg_{X \to x}^{(m)} \leq \delta). \tag{121}$$

By rearranging, $r \leq \delta + \left(\frac{\eta}{\Pr(lg_{X \to x}^{(m)} \leq \delta)}\right)^{1/p}$. Taking the infimum over $\delta$, we have the desired inequality. Again, we can apply the same idea for $s_\eta(x)$. $\square$

# F DATASET STATISTICS

We provide the statistics of the datasets including the number of data points, the number of features, the minimum and maximum values of the target value and the approximated Lipschitz constant in Table 2. The Lipschitz constant here is approximated by calculating the proportion $\frac{|y-y'|}{\|x-x'\|}$ for all pairs of data points then the value is given by the 95th percentiles of these proportions. We perform this procedure to avoid the outliers which have a size of around two orders of magnitude bigger than the 95th percentile value (Figure 4). This allows us to approximate the level of smoothness that does appear in the dataset rather than use the maximum Lipschitz constant. One could also think of this as a probabilistic Lipschitz value rather than the classical notion (Urner & Ben-David, 2013).

| Dataset | # data points | # features | [y min, y max] | Lipschitz constant |
|---------|---------------|------------|----------------|--------------------|
| Abalone | 4177 | 10 | [1,29] | 3.23 |
| Airfoil | 1503 | 5 | [103, 141] | 7.75 |
| Concrete | 1030 | 8 | [2,83] | 13.8 |
| Housing | 414 | 6 | [7, 118] | 11.68 |
| Power plant | 9568 | 4 | [420,496] | 14.18 |

Table 2: Dataset statistics.

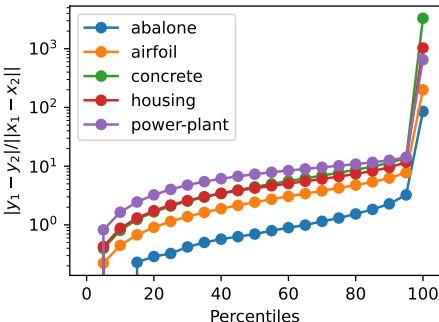

Figure 4: The value of $\frac{|y-y'|}{\|x-x'\|}$ by percentiles. We use the 95th percentile of this value as an approximated Lipschitz constant for each dataset.

# G IMPACTS OF THE INTERVAL SIZE AND INTERVAL LOCATION

## G.1 IMPACT OF THE INTERVAL SIZE

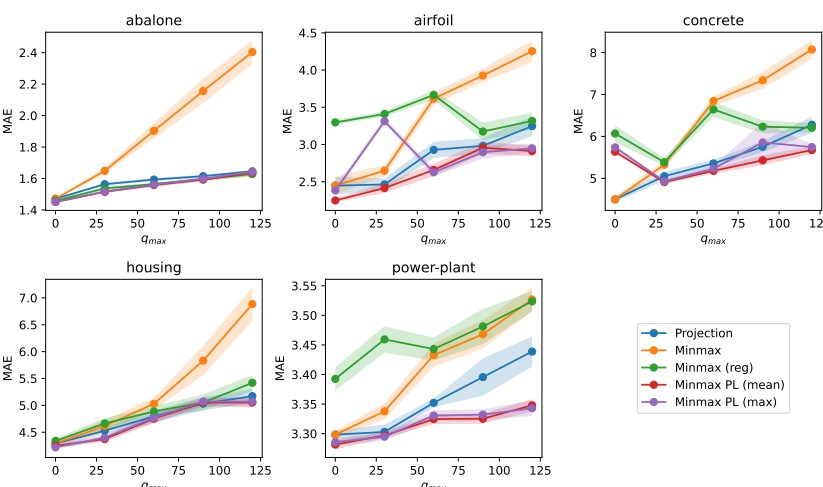

Figure 5: Test MAE when varying the maximum interval size $q_{max} \in \{0, 30, 60, 90, 120\}$ while $q_{min} = 0$.

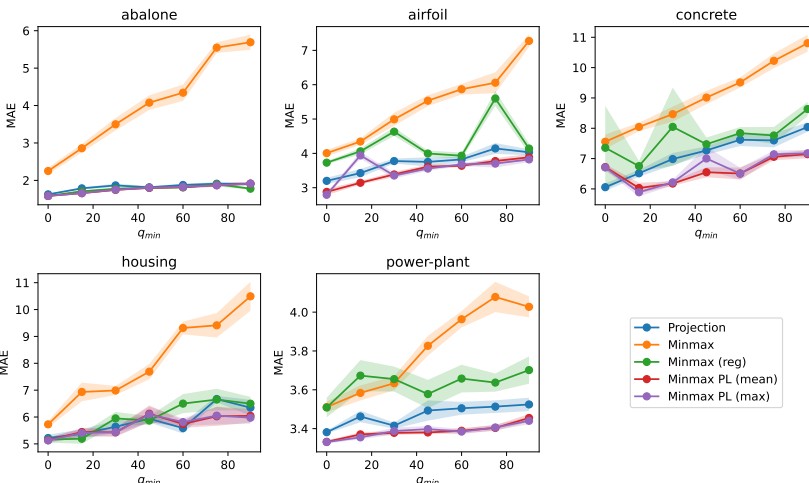

Figure 6: Test MAE when varying the minimum interval size $q_{min} \in \{0, 15, 30, 45, 60, 75, 90\}$ while $q_{max} = 90$.

We want to investigate the impact of interval size on the performance of the proposed methods. Intuitively, a smaller interval would make the problem easier. In the extreme case when the interval size is zero, we recover the supervised learning setting. Here, we assume that the interval location $p$ is still drawn uniformly from $[0, 1]$ and we consider two experiments. First, we vary the maximum interval size $q_{max} \in \{0, 30, 60, 90, 120\}$ while keeping the minimum interval size $q_{min} = 0$. As expected, a larger maximum interval size leads to the drop in test performance across the boards (Figure 5). Second, we vary the minimum inter val size $q_{min} \in \{0, 15, 30, 45, 60, 75, 90\}$ while keeping $q_{max}$ fixed at 90. We can see that the test performance also decreases for all methods as we increase the minimum interval size (Figure 6). Notably, the standard minmax approach is highly sensitive to the interval size where its performance degrades significantly much more than other approaches in both experiments. This is due to the nature of the approach that wants to minimize

the loss with respect to the worst-case label, as we have a larger interval, these worst-case labels can be much stronger and may not represent the property of the true labels anymore. On the other hand, our other minmax approaches and the projection approach are more robust to the change in the minimum interval size and the error only went up slightly for both experiments.

### G.2 IMPACT OF THE INTERVAL LOCATION

#### G.2.1 WHEN $y$ IS MORE LIKELY TO BE ON ONE SIDE OF THE INTERVAL (VARY $p_{\min}$)

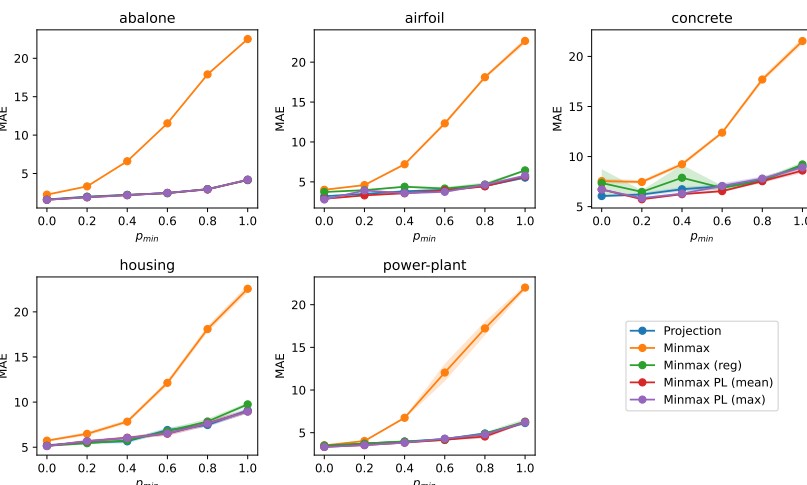

Figure 7: Test MAE when varying the minimum interval location $p_{\min} \in \{0, 0.2, 0.4, 0.6, 0.8, 1\}$. In this case, when $p_{\min} = 0$ we have the uniform interval setting while when $p_{\min} = 1$, $y$ true always lie on the upper bound of the intervals.

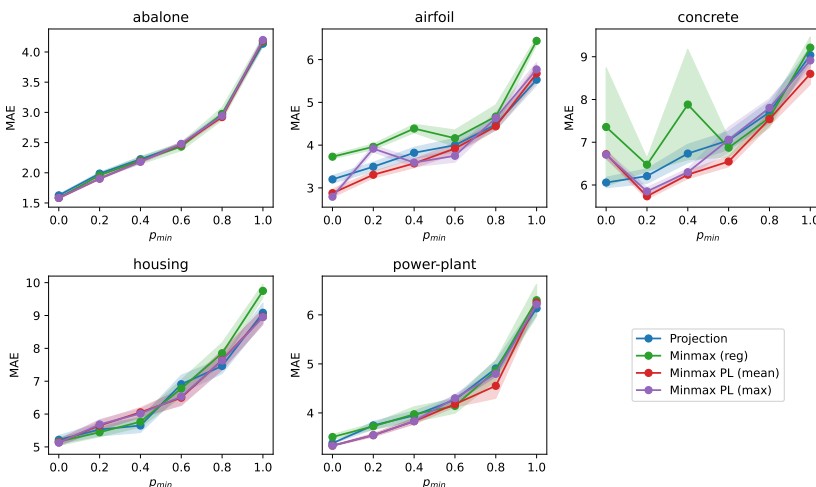

Figure 8: Test MAE when varying the minimum interval location $p_{\min} \in \{0, 0.2, 0.4, 0.6, 0.8, 1\}$. In this case, when $p_{\min} = 0$ we have the uniform interval setting while when $p_{\min} = 1$, $y$ true always lies on the upper bound of the intervals.(no minmax approach)

In the previous settings, we assume that the location of the interval $p$ is drawn uniformly from $U[0, 1]$, that is, when $y$ true is equally likely to be located at anywhere on the intervals. Here, we explore what would happen when it is not the case. We assume that we fixed $q_{\min} = 0, q_{\max} = 90$ and consider three scenarios. First, we consider when $y$ is more likely to be on one side of the

interval. Here, we consider when $p \sim U[p_{\min}, 1]$ where $p_{\min} \in \{0, 0.2, 0.4, 0.6, 0.8, 1\}$ (Figure 7). In this case, when $p_{\min} = 0$ we have the uniform interval setting while when $p_{\min} = 1$, $y$ true always lies on the upper bound of the intervals. We can see that the test MAE of all approaches increases as $p_{\min}$ is larger. Again, the minmax approach performs much worse than others. One explanation for this is that the minmax with respect to. the label would encourage the model to be close to the middle point of each interval (Corollary 4.2). However, the the $y$ true is far away from the midpoint leads to his phenomenon. We also provide the test MAE with no minmax approach for better visualization (Figure 8)

### G.2.2 When $y$ true is more likely to be in the middle of the interval

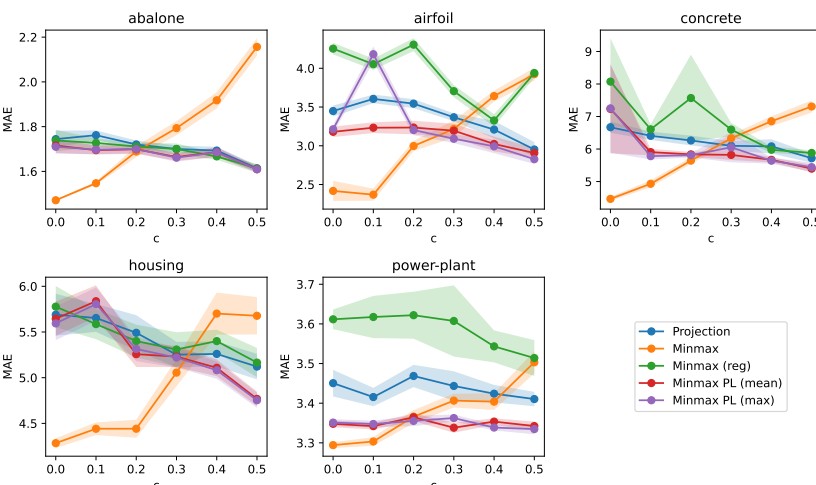

Figure 9: Test MAE when varying the interval location, $p \sim U[0.5 - c, 0.5 + c]$ for $c \in \{0, 0.1, 0.2, 0.3, 0.4, 0.5\}$. When $c = 0$, the true $y$ is always in the middle of the interval and when $c = 0.5$, we recover the uniform interval setting.

Second, we consider when $y$ true is more likely to be in the middle of the interval ($p$ is close to $0.5$). We capture this setting by considering $p \sim U[0.5 - c, 0.5 + c]$ for $c \in \{0, 0.1, 0.2, 0.3, 0.4, 0.5\}$ (Figure 9). Intuitively, when $c = 0$, the true $y$ is always in the middle of the interval and when $c = 0.5$, we recover the uniform interval setting. In contrast to the first setting, we can see that the minmax approach performs the best in this setting for a small value of $c$. Again, this is perhaps due to the nature of the minmax approach mentioned earlier which encourages the prediction to be close to the middle point of the interval, for which, in this case, close to the $y$ true. Remarkably, minmax performs better until $c = 0.2$ which corresponds to $p \sim [0.3, 0.7]$ which is a reasonable location of $y$ true in practice. However, when $c$ is large we would recover the uniform interval setting and the minmax would go back to becoming the worst-performer. On the other hand, the performance of other approaches is better as $c$ is larger, that is when $y$ true is more spread out across the interval.

### G.2.3 WHEN $y$ IS MORE LIKELY TO BE ON EITHER SIDE OF THE INTERVAL

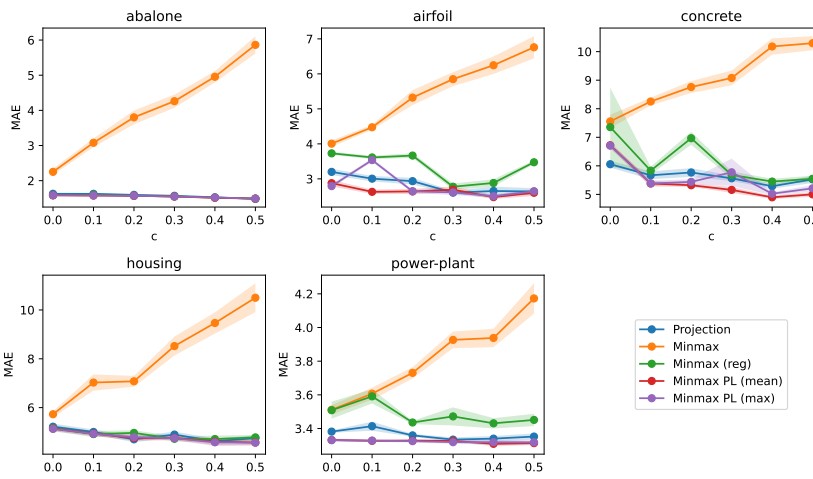

Figure 10: Test MAE when varying the interval location, when $p$ is drawn uniformly from $[0, 0.5 - c] \cup [0.5 + c, 1]$ when $c \in \{0, 0.1, 0.2, 0.3, 0.4, 0.5\}$. Here, when $c = 0$ we have the uniform interval setting while when $c = 0.5$, $y$ true is either on the upper or the lower bound of the intervals.

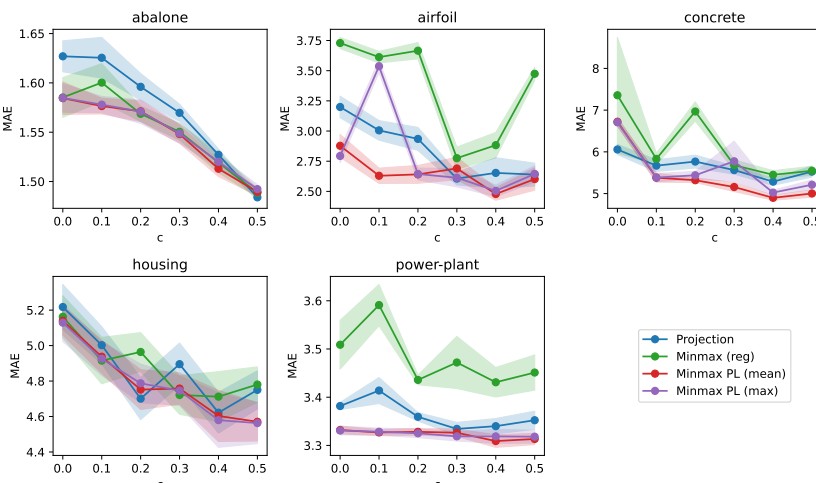

Figure 11: Test MAE when varying the interval location, when $p$ is drawn uniformly from $[0, 0.5 - c] \cup [0.5 + c, 1]$ when $c \in \{0, 0.1, 0.2, 0.3, 0.4, 0.5\}$. Here, when $c = 0$ we have the uniform interval setting while when $c = 0.5$, $y$ true is either on the upper or the lower bound of the intervals.(no minmax approach)

Finally, we consider when $y$ is more likely to be on either side of the interval where $p$ is drawn uniformly from $[0, 0.5 - c] \cup [0.5 + c, 1]$ when $c \in \{0, 0.1, 0.2, 0.3, 0.4, 0.5\}$. Here, when $c = 0$ we have the uniform interval setting while when $c = 0.5$, $y$ true is either on the upper or the lower bound of the intervals. We found that as $c$ is larger where the $y$ true is more likely to be near either of the boundaries, the minmax performance drop significantly (Figure 10). However, we found that the performance of other approaches increases (Figure 11). This is in contrast to the first setting where we see that when $y$ is more likely to be near only one side of the boundary, the performance drops remarkably.

Overall, from these experiments, we may conclude that for all approaches apart from the original minmax with respect to. labels, having $y$ true that lies near both of the boundaries of the interval are beneficial to the test performance and lying on both sides is crucial.

### G.3 LARGE AMBIGUITY DEGREE SETTING

We consider a setting with large ambiguity degree where $q \sim \text{Uniform}[q_{\min}, 90]$ when $q_{\min} \in \{30, 60, 90\}$ and $p \sim \text{Uniform}[0.5 - c, 0.5 + c]$ when $c \in \{0, 0.1, 0.2, 0.3, 0.4, 0.5\}$. Here as $c$ is smaller, $y$ true would be located near the middle point of the interval while as $c$ is larger, we would recover the uniform setting. These settings have a large ambiguity degree since when $q_{\min} > 0$, interval size can't be arbitrarily small and $[p_{\min}, p_{\max}] \subset [0, 1]$ implies that true y would not lie at the boundary of the constructed interval. As a result, the intersection of all possible intervals would no longer be just $\{y\}$ anymore which leads to the ambiguity degree of $1$. We found that there is no single method that always performs well on every interval setting. The Minmax is the best performing method for all $c \leq 0.3$ while when $c > 0.3$ the best-performing approaches are either PL (mean) or PL (max) (Figure 12).

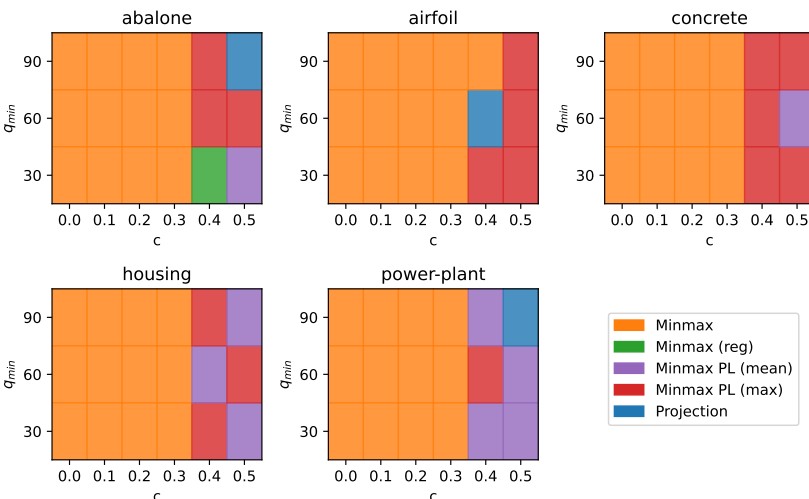

Figure 12: The best performing approach for each $c$ and $q_{\min}$

### G.4 INTERVAL PADDING EXPERIMENT

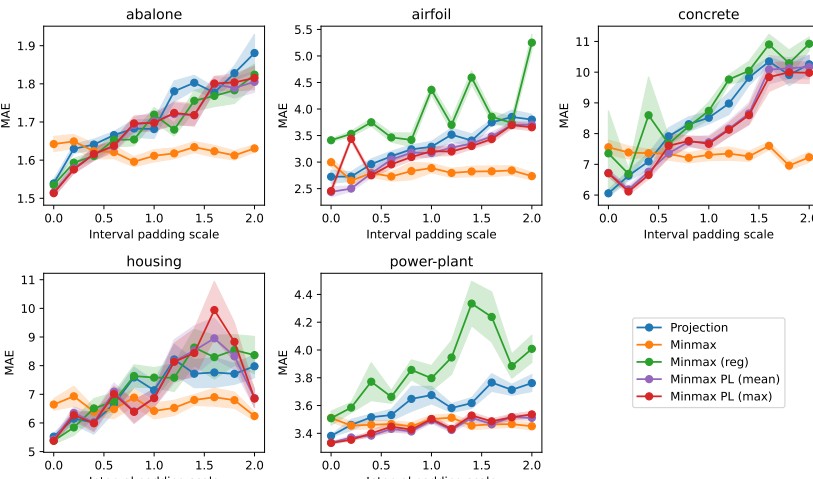

Figure 13: Interval padding experiment

From above, we found that the Minmax approach performs better when $y$ true is close to the middle point of the interval, but performs worse in the uniform interval setting when $p \sim \mathrm{Uniform}[0, 1]$. In this experiment, we start with the uniform interval setting and add padding to the original interval as a factor of the interval size. Formally, for an original interval $[l, u]$ of size $q = u - l$, we have a new interval $[l - sq, u + sq]$ when $s > 0$ is a scale parameter. By doing this, $y$ true would be *proportionally* closer to the midpoint of the new interval, but distancewise is the same. We found that as we add the padding, the performance of other approaches decreases significantly and gets worse than the performance of the Minmax when the scale is $0.5$ (when the padded interval is twice the size of the original interval) while the performance of Minmax is about the same. This shows that a redundant interval (padding) can harm the performance of the proposed approaches except Minmax and our result that interval location $p$ can have a large impact on the performance is still applicable to this padding setting.

# H INTERVAL SIZE AND TEST PERFORMANCE OF LIPMLP

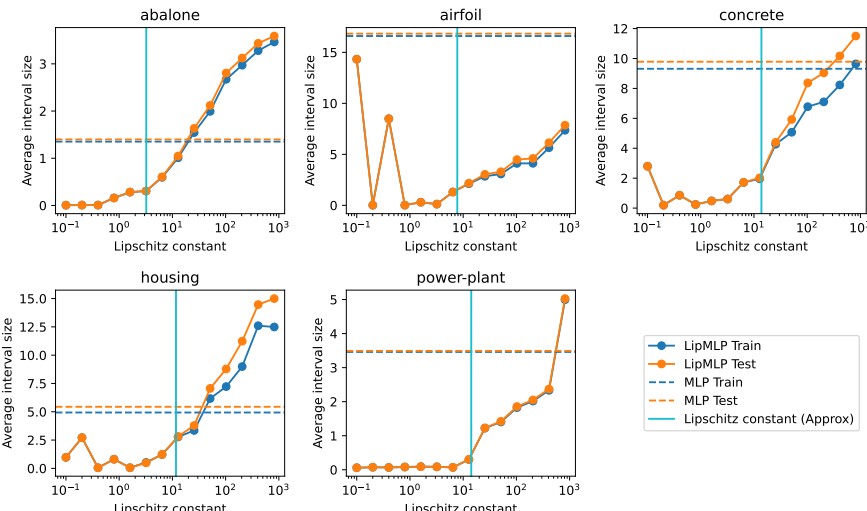

Figure 14: Approximated interval size $I_\eta(x)$ for Lipschitz MLP with a different value of Lipschitz constant $m$. The dashed horizontal lines are the values from standard MLP.

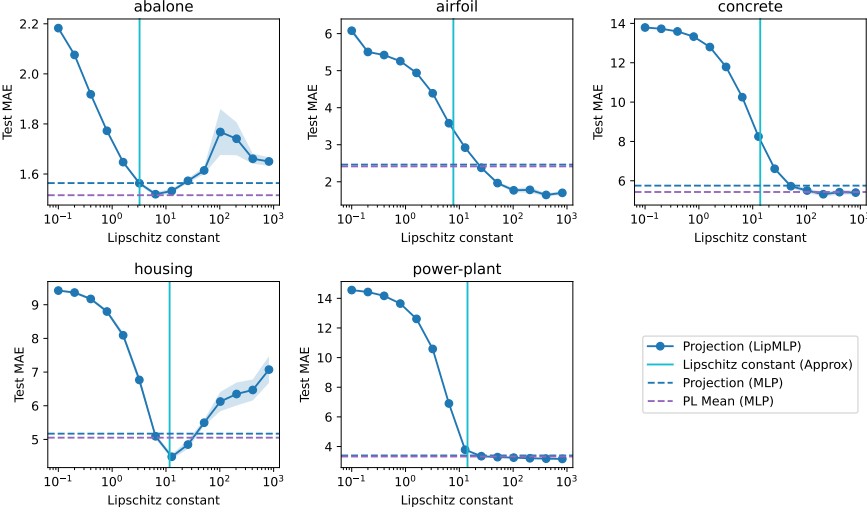

Figure 15: Test MAE of the projection method with Lipschitz MLP with different values of Lipschitz constant. The vertical line is the Lipschitz constant approximated from the training set. The dashed horizontal lines are the test MAE of PL (Mean) and Projection approach with a standard MLP.

# I   ABLATION FOR PL (MEAN)

Since PL (mean) is the best-performing approach in the uniform interval setting, we also performed an ablation study to improve our understanding of this method. First, we explore the impact of the number of hypotheses $k$ used to represent $\widetilde{\mathcal{F}}_0$. We found that for every dataset, as $k$ is larger, the test MAE becomes smaller. While we use $k = 5$ for all PL experiments, this ablation suggests that we can increase $k$ to get better performance at the cost of more computation.

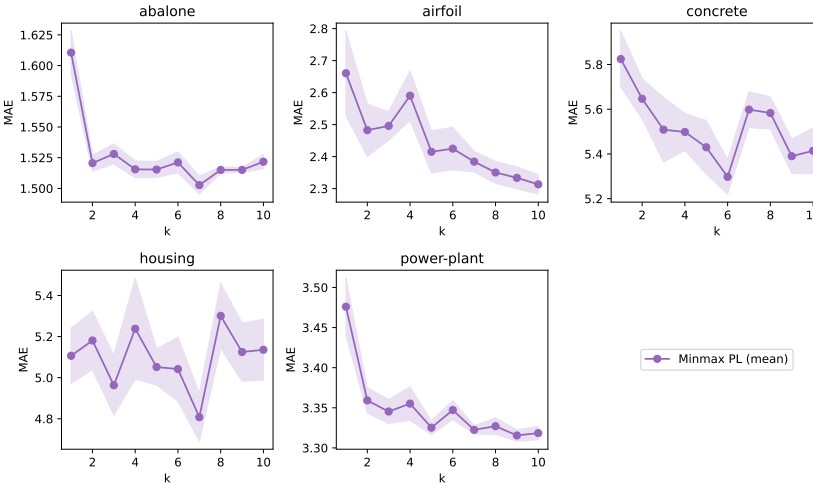

Figure 16: Test MAE for PL (mean) with different number of hypotheses $k$ used to represent $\widetilde{\mathcal{F}}_0$. For almost every dataset, the test MAE decreases as $k$ is larger.

Second, we also compare PL (mean) with a natural ensemble baseline where we combine pseudo labels by averaging them first and then train a model with respect to. the averaged labels. In particular, the objective for the ensemble baseline is given by

$$\min_f \sum_{i=1}^{n} \ell(f(x_i), \sum_{j=1}^{k} f_j(x_i)). \tag{122}$$

We found that PL (mean) still performs better than this baseline on 2 out of 5 datasets while the other 3 datasets are similar.

|  | Abalone | Airfoil | Concrete | Housing | Power-plant |
|---|---|---|---|---|---|
| PL (mean) | $1.52_{0.01}$ | $2.42_{0.07}$ | $5.43_{0.12}$ | $5.05_{0.09}$ | $3.33_{0.01}$ |
| PL ensemble baseline | $1.51_{0.01}$ | $3.3_{0.04}$ | $5.57_{0.19}$ | $5.06_{0.08}$ | $3.32_{0.01}$ |

Table 3: Test Mean Absolute Error (MAE) and the standard error (over 10 random seeds) for PL (Mean) and a PL ensemble baseline

