# OpenReview forum: "Learning from interval targets"
_ICLR.cc/2025/Conference — Submitted to ICLR 2025_

### Official Review · Reviewer_9hQV · 2024-10-31

**Soundness:** 3
**Presentation:** 3
**Contribution:** 3
**Rating:** 6
**Confidence:** 2

**Summary:**

This paper studies a regression model where only an interval around the true response is known to the learner. Given such weak supervision, the goal of the learner is to learn a predictor whose population risk is small. To tackle this problem, the authors propose two methods. The first modifies the standard regression loss functions to account for interval ground truths. In particular, a projection loss is defined which penalizes the the output a learned predictor when its outputs lie outside the interval for a specific instance. The second formulates a minimax problem with an objective that minimizes the original regression loss with respect to the worst-case label within the interval (available to the learner). In both settings, the authors show how the smoothness of the function class (measured via Lipschitzness) can be beneficial. The authors provide theoretical guarantees for both methods and corroborate their results with experiments.

**Strengths:**

- Paper is well written and organized
- Paper improves upon previous works by removing assumptions and the ideas seem novel as far as I can tell
- I think the Figures do a good job of giving intuition about the methods

**Weaknesses:**

- Unclear objective. I think the authors can do a better job about making explicit the fact that although the learner observes interval targets, the goal is to still do well with respect to the original regression loss. This goal is sort of hidden in lines 101-102, and it would be nice to emphasize this more.

- Lack of specification on how error rates decay with sample size. It is not clear how the excess risk in lines (13) and (16) decays with the sample size $n$. For all I know, it may not decay at all, making the error bounds vacuous. At the very minimum, it would be nice if the authors can provide error rates for a specific class of functions. This is also an issue in line (22) - I have no intuition on how the error might decay with the sample size here.  I think this is a significant limitation and is the main reason for my rating. I will raise my score if the authors can give me a sense of how the non-asymptotic error rates decay with $n$, even it is for a specific function class.

- Unclear method. I am confused about certain aspects of the methods proposed in Section 4.1 See Questions below for more details.

**Questions:**

- Does Proposition 3.1 hold uniformly for all $x \in X$? If so, I am confused since $f \in \tilde{F_{0}}$ only guarantees that $f(x) \in [l_x, u_x]$ almost surely. So there could still be $x$'s (albeit in a measure $0$ set), for which this is not true.
- What norm is being used on line 236?
- Is the guarantee on line 267 pointwise or in expectation? If it is pointwise for every $x$, how are you accounting for the fact that the functions $f \in \tilde{F_{\eta}}$ may have projection loss bigger than $\eta$ for low probability $x$? Is this accounted for by definition of $r_{\eta}(x)$ and $s_{\eta}(x)$? If so, it would be good to explain how the buffer $r_{\eta}(x)$ and $s_{\eta}(x)$ account for that.
- Is $\tilde{F}_0$ known to the learner in Equation (23)? It seems like the learner needs to know the distribution $D$ over $X$ to compute $\tilde{F}_0$ based on line 170. So because the learner doesn't know $D$, it shouldn't be able to perform the optimization in Equation (23). What am I missing here?
- Based on the bullet points above, how does the learner know that the $f_1, ..., f_j$ in line 409-410 belong to $\tilde{\mathcal{F}}_0$ if $\tilde{\mathcal{F}}_0$ is not known?
- Do the upper bounds in Equations (13) and (16) decay with the sample size $n$? If so, it would be good to comment and be explicit about how they decay with $n$ (i.e what is the rate of decay?).
- In Equation (24), are the minimizations and maximizations over $f$ and $f^{\prime}$ being done both over $\mathcal{F}$? Or is the maximization of $f^{\prime}$ being done over $\tilde{F}_0$? If it is the latter, the same question I had above holds - how can the learner do this if $\tilde{F}_0$ is unknown?

---

> ### Author Response · Authors · 2024-11-20
>
> Thank you for your feedback and suggestions. We hope the following responses address all your questions.
>
> > 1. I think the authors can do a better job about making explicit the fact that although the learner observes interval targets, the goal is to still do well with respect to the original regression loss.
>
> We have made this setting more explicit in the revised version.
>
>
>
> > 2. Lack of specification on how error rates decay with sample size. It is not clear how the excess risk in lines (13) and (16) decays with the sample size n.
>
> We have added an explicit generalization bound for both realizable (13) and agnostic setting (16) in terms of the number of data points $n$ in Appendix A. This result is applicable to any hypothesis class $\mathcal{F}$ where the Rademacher complexity grows as $O(1/\sqrt{n})$. For instance, the class of two-layer neural network with bounded weight satisfies this property.
>
>
> > 3. Does Proposition 3.1 hold uniformly for all $x \in X$? If so, I am confused since $f \in \mathcal{F}_0$ only guarantees that $f(x) \in [l_x, u_x]$ almost surely. So there could still be $x$ (albeit in a measure 0 set), for which this is not true.
>
> Our result in Proposition 3.1 holds for any $x \in \mathcal{X}$ with a positive probability density function $p(x) > 0$. We have made this more clear in the revised version. We note that this is stronger than $f(x) \in [l_x, u_x]$ almost surely, and we deal with a set with measure 0 with an assumption that the distribution $D_I$ is a non-atomic distribution (Assumption 2 in Appendix E).
>
>
> > 4. What norm is being used on line 236?
>
> We used $L_1$ norm and this is made clearer in the revision.
>
>
> > 5. Is the guarantee on line 267 pointwise or in expectation? If it is pointwise for every $x$, how are you accounting for the fact that the functions $f \in F_\eta $ may have projection loss bigger than $\eta$ for low probability x? Is this accounted for by definition of  $r_{\eta}(x)$ and $s_{\eta}(x)$? If so, it would be good to explain how the buffer $r_{\eta}(x)$ and $s_{\eta}(x)$ account for that.
>
> The guarantee in line 267, is **pointwise** and holds for any $x \in \mathcal{X}$ with a positive probability density $p(x) > 0$. We are able to deal with any $x$ with a large projection loss (although with low probability) by relating $x$ with its neighbor $x'$. The main idea is that, with smoothness, if $f(x)$ is too small (too large) then for any neighbor $x'$, $f(x')$ has to be close to $f(x)$ and thus, would be too small (too large) as well. But this can't happen too much since the expected projection loss is smaller than $\eta$ and we bound how much $f(x)$ can be far away from the given interval with $r_{\eta}(x)$ and $s_{\eta}(x)$. The full proof in Appendix C. We will make this connection clearer in the revised version.
>
>
> > 6. Is $F_0$ known to the learner in Equation (23)? It seems like the learner needs to know the distribution D over X to compute  $F_0$ based on line 170. So because the learner doesn't know $\mathcal{D}$, it shouldn't be able to perform the optimization in Equation (23). What am I missing here?
>
>
> Yes, we would not have access to $F_0$ in practice. Therefore, we propose to approximate the objective in equation (23) with the regularization approach and with pseudo labels. In general, we could learn $f_j \in F_\eta$ for some small $\eta$ and we could treat a set of such functions as an approximation of $F_0$ (pseudo labels idea). We have made this connection more explicit in the revised version.
>
>
> > 7. Based on the bullet points above, how does the learner know that the $f_1,...,f_j$ in line 409-410 belong to $F_0$ if $F_0$  is not known?
>
> In practice, $f_1,...,f_j$ would belong to $F_{\eta}$ for some small $\eta$. We can achieve this by training with a different random seed. $F_\eta$ is a superset of $F_0$ where we use it to approximate $F_0$. We have made this approximation aspect is more clear in the revised version.
>
>
>
>
> > 8. In Equation (24), are the minimizations and maximizations over $f$ and $f'$ being done both over $\mathcal{F}$? Or is the maximization of $f'$ being done over $\mathcal{F}_0$? If it is the latter, the same question I had above holds - how can the learner do this if $\mathcal{F}_0$ is unknown?
>
> The minimizations and maximizations over $f$ and $f'$ are being done both over $\mathcal{F}$. However, we have a regularization on $f'$ to encourage $f'$ to be in $\mathcal{F}_0$.

---

> > ### Comment · Reviewer_9hQV · 2024-11-23
> >
> > I thank the authors for their response. As promised, I have increased my score to 6 given their explicit generalization bounds.

---

### Official Review · Reviewer_UKDx · 2024-10-31

**Soundness:** 3
**Presentation:** 3
**Contribution:** 4
**Rating:** 6
**Confidence:** 3

**Summary:**

This paper considers a regression analysis problem where the outcome of each training sample is given by not single value but an interval. An existing method assumes the "realizability" and "ambiguity degree < 1" in the existing method, and it employs the loss function called the "projection loss" and proved the theoretical performances. However, since these assumptions are too restrictive, the paper relaxes the assumptions, and accordingly constructed another learning setup called "minmax objective".

**Strengths:**

-   The formulations of the problem and the method looks clear.
-   The method introduced less restrictive constraint than existing work: the class of hypotheses ${\cal F}$ must contain only $m$-Lipschitz functions. Also, the experiment in Section 5.2 examines the effect of the choice of $m$ in the algorithm. (As discussed with equation (16), $m$ should not be too small or too large.)

**Weaknesses:**

-   It looks that a part of experimental setups are lacked. Please see Section "Question" below.
-   Just notation problems, but it is difficult to follow the effect of the Lipschitz constant $m$. It may be better to add "$m$" to all operations affected by $m$ (e.g., replacing $l_{x^\prime\to x}$ with $l^{(m)}_{x^\prime\to x}$).

**Questions:**

-   Section 2, line 133: What is the "$\ell$1 loss as the surrogate" for the 0-1 loss? The l1-loss surrogate for (1) is not so obvious, although we can understand it after reading its mathematical expression in (Cheng et al., 2023a).
-   Section 2, equation (2): What are the capital letters $L$ and $U$? Perhaps the lower and upper bounds as random variables? (If so, what are their distributions?)
-   Section 3, equation (6) and the succeeding sentences: Are $\pi_{\ell}$ and $\pi_l$ the same? (Perhaps just non-unified notations?)
-   Section 3.2, Proposition 3.2: What situation the induced lower bound and upper bounds of $f(x)$ is effective? As far as my understanding, if the size of the bounds is small at $x^\prime$, the size at another point $x$ must be accordingly small under the constraint of Lipschitz continuity.
-   Section 3.3.1, equation (13): What are the definitions of $I_0$ and $I_\eta$? (Although I could guess them,) their specific definitions seems not to be given.
-   Section 3.3.1, equation (14): Is the equation correct? The rightmost expression of the equation looks like independent from $x$.
-   Section 4: Are the notations "$f^\prime$"'s in equations (20) and (21) are related? If not, it is better to introduce different variables.
-   Section 5 (Appendix D): What is the obstacle (e.g., computational cost) to run the proposed method for a larger dataset? The datasets used in the experiments have up to 10000 samples.
-   Section 5: The proposed method assumes that the class of hypotheses is $m$-Lipschitz, but in the experiment how can we limit the prediction model (MLP in this case) as $m$-Lipschitz? (The paper refers the existing work (Cheng et al., 2023a), but in the existing work it looks that $m$-Lipschitz is not employed.)
-   Section 5: What loss function does the experiment employed? (More specifically, $\psi$ in Assumption 1)

---

> ### Author Response · Authors · 2024-11-20
>
> Thank you for your feedback and questions. We hope the following responses address all your questions.
>
> > 1. Just notation problems, but it is difficult to follow the effect of the Lipschitz constant m. It may be better to add m to all operations affected by m (e.g., replacing $l_{x' \to x}$ with $l_{x' \to x}^{(m)}$).
>
> We have incorporated this in the revised version.
>
>
> > 2. Section 2, line 133: What is the "$\ell_1$ loss as the surrogate" for the 0-1 loss? The l1-loss surrogate for (1) is not so obvious, although we can understand it after reading its mathematical expression in (Cheng et al., 2023a).
>
> Yes, by $\ell_1$ surrogate loss for 0-1 loss, we refer to equation 12) in Cheng et al. 2023a. We made this more explicit in the revised version.
>
>
> > 3. Section 2, equation (2): What are the capital letters L and U ? Perhaps the lower and upper bounds as random variables? (If so, what are their distributions?)
>
> Yes, L and U are the lower and upper bounds as random variables. We assume that the tuple $(x_i, l_i, u_i)$ is sampled i.i.d. from a distribution $\mathcal{D}_I$ (line 106).
>
>
> > 4. Section 3, equation (6) and the succeeding sentences: Are $\pi_\ell$ and $\pi_l$ the same? (Perhaps just non-unified notations?)
>
> Yes, we have fixed this in the revised version.
>
>
> > 5. Section 3.2, Proposition 3.2: What situation the induced lower bound and upper bounds of $f(x)$ is effective? As far as my understanding, if the size of the bounds is small at  $X'$ ,the size at another point $x$ must be accordingly small under the constraint of Lipschitz continuity.
>
> First, we note that under a small ambiguity degree assumption, our bound would be very effective where we would recover the true label, $[l_{\mathcal{D} \to x}, u_{\mathcal{D} \to x}] = \{y\}$. Our bound is also applicable when the ambiguity degree is large  and yes, you have the right intuition. Our bound is more effective when the hypothesis class is smoother, and when the neighbor intervals are "far away'' from the given interval, so that it would force the hypothesis to be near the edge of the given interval (see Figure 1b for an illustration).
>
>
> > 6. Section 3.3.1, equation (13): What are the definitions of $I_0$ and $I_\eta$ ? (Although I could guess them,) their specific definitions seems not to be given.
>
> We had defined $I_\eta$ in Theorem 3.3 in the previous version. In the revision, we define this explicitly in Section 3.3 so that it is easier for the reader.
>
>
>
> > 7. Section 3.3.1, equation (14): Is the equation correct? The rightmost expression of the equation looks like independent from $x$.
>
> Yes, this is correct. Here we consider and example when our error bound is tight when the hypothesis class only contains a constant function. Therefore, the right-hand side is independent from $x$.
>
>
>
> > 8. Section 4: Are the notations $f$ in equations (20) and (21) are related? If not, it is better to introduce different variables.
>
> Yes, they are the same hypothesis $f$ that we learn from the hypothesis class $\mathcal{F}$. However, they are not related and we can incoporate this in the revised version.
>
>
> > 9. Section 5 (Appendix D): What is the obstacle (e.g., computational cost) to run the proposed method for a larger dataset? The datasets used in the experiments have up to 10000 samples.
>
>
> There is no bottleneck for running this method on a larger dataset. In particular, the complexity of the projection method is the same as the complexity of traditional regression (with the same function class). For the pseudo-label methods, the complexity is $k$ times of the complexity of the traditional regression method when $k$ is the number of pseudo labels for each instance. We expect our proposed methods to scale  well with the size of the datasets.
>
>
>
> > 10. Section 5: The proposed method assumes that the class of hypotheses is m-Lipschitz, but in the experiment how can we limit the prediction model (MLP in this case) as m-Lipschitz? (The paper refers the existing work (Cheng et al., 2023a), but in the existing work it looks that m-Lipschitz is not employed.)
>
> We use an MLP with a spectral normalization layer (Miyato et al. 2018)(see Section 5.2). The normalization ensures that the Lipschitz constant of the MLP is less than 1. We then scale the output of the MLP by a constant factor m to ensure that the Lipschitz constant of the hypothesis is less than $m$.
>
> > 11. Section 5: What loss function does the experiment employed? (More specifically, $\psi$ in Assumption 1)
>
> We use mean absolute error or $l_1$ loss in our experiment. This is equivalent to $\psi(x) = x$ in Assumption 1.

---

> ### Author Response · Authors · 2024-11-25
>
> Dear reviewer, please let us know if you have any other questions. If we have resolved all your concerns, would you be willing to reconsider your score?

---

> ### Comment · Reviewer_UKDx · 2024-11-26
>
> I was convinced with author's responses, however, I will retain my score after reading others' reviews.

---

### Official Review · Reviewer_giwV · 2024-11-01

**Soundness:** 3
**Presentation:** 3
**Contribution:** 3
**Rating:** 6
**Confidence:** 5

**Summary:**

This paper investigates a weakly supervised regression setting called “learning from interval targets," where the exact real-valued targets are not directly available; instead, supervision is provided in the form of intervals around the targets. The authors suggest that previous research assumptions for theoretical analysis were overly simplistic (SMALL AMBIGUITY DEGREE ASSUMPTIONS). Therefore, they propose new theoretical analyses to address more robust real-world scenarios. To tackle this problem, the authors propose two methods: (1) using surrogate loss functions to align targets with weak supervision labels (PROJECTION APPROACH); and (2) considering the worst-case scenario within the intervals and minimizing that risk (MINMAX OBJECTIVE). Extensive experiments validate the effectiveness of the authors' methods.

**Strengths:**

- The authors discuss an intriguing weakly supervised regression problem: learning from interval targets. Most research has only considered classification problems, but they overlook that regression problems are equally important.
- The authors provide (reasonable) theoretical analyses for their methods, demonstrating the effectiveness of the proposed approaches.
- The methods proposed by the authors are both natural and reasonable, and the paper is well-organized and easy to understand.

**Weaknesses:**

- The experimental setup considered by the authors is too simplistic, as it only involves a small number of UCI datasets, and the selected datasets are simple (small and low-dimensional). Previous research seems to have also considered experimental setups in CV and NLP.

- The authors' analysis is based on the rejection of the SMALL AMBIGUITY DEGREE ASSUMPTIONS, but I believe their rejection of this assumption is not convincing enough, as it would be impossible to learn if noise is always present.

**Questions:**

- The authors believe that the SMALL AMBIGUITY DEGREE ASSUMPTIONS are not applicable in regression scenarios primarily because regression tasks differ from classification tasks in having exact labels. Therefore, directly applying the previous SMALL AMBIGUITY DEGREE ASSUMPTIONS is unreasonable. Should the AMBIGUITY DEGREE ASSUMPTIONS be redefined instead of being directly rejected? A larger AMBIGUITY DEGREE might result in intervals that are always mixed with noise, making it impossible to discern the true label distribution. Alternatively, could the size of intervals be used to constrain the AMBIGUITY DEGREE to satisfy a new relationship?
- In the PL (max) and PL (mean) methods, the authors utilize pseudo labels, but it seems there is no mention of how these pseudo labels are generated.
- The projection loss proposed by the authors (Equation 5) appears to be consistent with the surrogate loss function in [1]. Does this imply that the method in [1] can still be applicable under the new setting (SMALL AMBIGUITY DEGREE ASSUMPTIONS does not apply) ?

---

> ### Author Response · Authors · 2024-11-20
>
> Thank you for your feedback and suggestions. We hope that the following address your main questions regarding the ambiguity degree assumption.
>
> > 1. The experimental setup considered by the authors is too simplistic, as it only involves a small number of UCI datasets, and the selected datasets are simple (small and low-dimensional). Previous research seems to have also considered experimental setups in CV and NLP.
>
> Our focus on a small set of UCI datasets aims to complement the theoretical contributions of this work with clear, reproducible results. While it would be very valuable, our focus was not on a large-scale and thorough evaluation on the experimental front.
>
>
> > 2. The authors' analysis is based on the rejection of the SMALL AMBIGUITY DEGREE ASSUMPTIONS, but I believe their rejection of this assumption is not convincing enough, as it would be impossible to learn if noise is always present.
>
> In regression, predictions close to the target (within an error tolerance $\epsilon$) are often sufficient, making learning with noise more tolerable. For example, when for each label $y$ the given interval is a ball with radius $\epsilon > 0$ around $y$, $[l,u] = [y - \epsilon, y+\epsilon]$. When $\epsilon$ is small enough, then we would expect to be able to learn a good function, albeit with some irreducible (small) error that depends on $\epsilon$. However, since the ambiguity degree is large in this setting, the previous work would not apply.
>
> > 3. In the regression setting,) should the AMBIGUITY DEGREE ASSUMPTIONS be redefined instead of being directly rejected? A larger AMBIGUITY DEGREE might result in intervals that are always mixed with noise, making it impossible to discern the true label distribution. Alternatively, could the size of intervals be used to constrain the AMBIGUITY DEGREE to satisfy a new relationship?
>
> We have now added a note in the paper outlining an alternate definition in line with your proposal, and how it relates to our definition, in Appendix B. Here's a preview:
>
> Recall that the small ambiguity degree assumption implies that for each instance $x$, the intersection of all intervals of $x$ is the true label $y$; $\bigcap [l_x, u_x] = y$. One possibility along the lines of your suggestion is to redefine it so that the intersection falls within a radius $\epsilon$ around the true label: $\bigcap [l_x, u_x] \subseteq B(y, \epsilon)$. In this case, our analysis already captures the essence of this interval intersection for each $x$. If we have this assumption then our bound would already imply that $f(x) \in I_0(x) \subseteq B(y, \epsilon)$. Further, $I_0(x)$ might even be a proper subset of the ball $B(y, \epsilon)$ based on the smoothness property.
>
>
> > 4. In the PL (max) and PL (mean) methods, the authors utilize pseudo labels, but it seems there is no mention of how these pseudo labels are generated.
>
> The pseudo labels are generated from a hypothesis $f_j$ which is a function from $\widetilde{\mathcal{F}}_\eta$ (see equation (25), (26)) for some small $\eta$. We have updated the paper to make this more clear now.
>
> > 5. The projection loss proposed by the authors (Equation 5) appears to be consistent with the surrogate loss function in [1]. Does this imply that the method in [1] can still be applicable under the new setting (SMALL AMBIGUITY DEGREE ASSUMPTIONS does not apply) ?
>
> We suppose that [1] here refers to a prior work "Weakly supervised regression with interval targets" by Cheng et al., 2023a. The surrogate loss function in [1] is a special case of our $L_p$ loss when $p=1$. In our paper, we have a theoretical result for $L_p$ loss for all $p \geq 1$ and also for a general loss function that satisfies Assumption 1. Our general result implies that [1] is still applicable in the setting when the small ambiguity degree assumption does not apply. On the other hand, previous work does not use the smoothness assumption and do not tune the Lipschitz constant of the hypothesis class. Our analysis found that this is an important variable and demonstrate that tuning the Lipschitz constant leads to a better downstream performance (Figure 3).

---

> > ### Comment · Reviewer_giwV · 2024-11-21
> >
> > I thank the authors for their response, which has addressed some of my concerns. However, I still have a question regarding Question 4.
> >
> > While I understand that the pseudo labels are generated by the hypothesis $f_{j}$, it remains unclear how the hypothesis $f_{j}$ is obtained. The additional section mentions that $f_{j}$ is derived by minimizing the empirical projection loss, but this explanation is still insufficiently detailed. For example, what exactly constitutes the empirical projection loss? What data is used in this process? Specifically, the setup used in the experiments to determine $f_{j}$ is not clearly explained, which should be an important component of PL (max) and PL (mean) methods.

---

> ### Author Response · Authors · 2024-11-21
>
> We are happy to provide an additional clarification. The empirical projection is given by Equation (5). That is, if we have training samples $(x_i, l_i, u_i)$ for $i = 1,\dots, n$ , then
>
> $f_j \in \arg\min_{f \in \mathcal{F}} \frac{1}{n} \sum_{i=1}^n \pi_\ell(f(x_i), l_i, u_i).$
>
> In practice, training a model using stochastic gradient descent with different random initializations and random seeds ensures that $f_j$ will be different. This process can be repeated $k$ times to obtain $\{f_1, \dots, f_k\}$. We will revise this to be more explicit in the updated version.

---

> > ### Author Response · Authors · 2024-11-25
> >
> > Dear reviewer, please let us know if you have any other questions. If we have resolved all your concerns, would you be willing to reconsider your score?

---

> > > ### Comment · Reviewer_giwV · 2024-11-26
> > >
> > > Thank you to the authors for the additional clarification, which has resolved my confusion regarding hypothesis $f_{j}$. I will maintain my current score 6, which is a positive rating.

---

### Official Review · Reviewer_E2Ty · 2024-11-03

**Soundness:** 2
**Presentation:** 3
**Contribution:** 2
**Rating:** 5
**Confidence:** 4

**Summary:**

This paper investigates the regression problem in a situation where the learner has access only to an interval around the true response. It proposes two approaches:

1. Modifying the regression loss function with a projection and learning a hypothesis lying within the interval
2. Minimizing a typical regression loss with respect to the worst-case label

**Strengths:**

This work investigates an interesting situation where the learner has access only to an interval around the true response, rather than the exact response as in the standard supervised learning setting. Due to the high cost of getting exact responses, learning from intervals has significant practical meanings.

It proposes two approaches and provides both theoretical analysis and experiments on real-world datasets. For the theoretical results, the proofs are clear and easy to follow, accompanied by a sufficient explanation of the derivations.

**Weaknesses:**

1. This paper highlights a generalization bound in Section 3.3 as its main result. However, this bound seems too straightforward and can be quite vacuous -  e.g., Theorem 3.5 is only taking an expectation of Equation (12), which is the worst-case analysis on a single instance.
2. The multiple parameters appearing in the generalization bound are difficult, if impossible at all, to obtain. For example, in order to get the bound in Theorem 3.5, one needs to know $I_\eta$. $I_\eta$ consists $r_\eta$, which involves the calculation of an expectation with respect to $X$ (Equation 9). Usually, one is not able to compute the expectation efficiently.
3. To obtain an estimate of $\eta$, one needs to know the Rademacher complexity of $\Pi(F)$, which can be quite difficult to obtain.

**Questions:**

1. It's mentioned that Theorem 3.5 can be tight for certain hypothesis classes, for instance, constant hypotheses. However, the constant hypothesis seems to be too restrictive. Could you provide another instance of hypothesis class, where this theorem can be relatively tight?
2. Is it possible to obtain the parameters in the generalization bounds in an efficient way?
3. In lines 179-182, it's mentioned that a value of $\eta$ is obtained from the RHS of Equation (6); how would one calculate the Rademacher complexity of $\Pi(F)$? How does the tightness of an estimation of $\eta$ affect the later generalization bounds?
4. In lines 201-202, why $\eta_n = O(1/\sqrt n)$? How does the Rademacher complexity of $\Pi(F)$ affect the order of $\eta_n$?

---

> ### Author Response · Authors · 2024-11-20
>
> Thank you for your feedback and suggestions. We hope that the following address your questions regarding our theoretical results.
>
>
> > 1. This paper highlights a generalization bound in Section 3.3 as its main result. However, this bound seems too straightforward and can be quite vacuous.
>
>
> We want to highlight that one of the technical difficulties in proving Theorem 3.3 is that we allow for $f$ that only approximately lies inside the interval. That is, for certain $x$, $f(x)$ could be arbitrarily far away from the given interval $[l_x, u_x]$. Nevertheless, our bound provides a **pointwise** reduced interval for each individual $x$. This is non-trivial as we have to work with $f(x)$ that could be arbitrarily far away from the given interval $[l_x, u_x]$.
>
>
>
>   > 2.  The multiple parameters appearing in the generalization bound are difficult, if impossible at all, to obtain.
>     \ldots To obtain an estimate of $\eta$ one needs to know the Rademacher complexity of $\Pi(F)$ which can be quite difficult to obtain.
>
> Our bounds emphasize *learnability* in the agnostic setting, which was not known in previous work. That is, as the sample size increases, we can bring error down to the error rate for the best hypothesis in class. Our bounds also reveal that the Lipschitz constant affects generalization. **In practice, we would not estimate $I_\eta$ or $r_\eta$** but instead tune relevant parameters, and our analysis shows that the smoothness parameter is an important one to tune. Our experiments confirm the insights of the analysis hold in practice.
>
>
> > 3. It's mentioned that Theorem 3.5 can be tight for certain hypothesis classes, for instance, constant hypotheses
>     $\dots$ Could you provide another instance of hypothesis class, where this theorem can be relatively tight?
>
> The bound would also be relatively tight on both extreme when the hypothesis class is  smoother (Lipschitz constant $m$ is close to $0$) and when the hypothesis class is very flexible ($m$ is very large).  The first setting follows from the fact that the bound holds for a constant hypothesis class when $m = 0$. For the second example, when the hypothesis class is very flexible so that the label of any $x$, is independent of other $x'$, the only information we have is that the true label lies inside a given interval then the best thing we could hope for is that the error for each instance $x$ is at most the size of the interval of each $x$. Since the hypothesis class is too flexible, we can't derive additional information from it.
>
>
>
> > 4. In lines 179-182, it's mentioned that a value of $\eta$ is obtained from the RHS of Equation (6); how would one calculate the Rademacher complexity of  $\Pi(F)$ ? How does the tightness of an estimation of $\eta$ affect the later generalization bounds?
> $\dots$ In lines 201-202, why $\eta_n=O(1/\sqrt{n})$? How does the Rademacher complexity of $\Pi(F)$ affect the order of $\eta_n$?
>
>
> The result in the line 179-182 is an upper bound of the true value of $\eta$ of the learned hypothesis $f$ (denoted as $\eta_n$). This bound is for a theoretical purpose where we hope to communicate that as we have more training data points $n$, $\eta$ would converge to zero as $\eta_n$ is of $O(1/\sqrt{n})$. As a result,  we would be able to learn a hypothesis that lies inside the given intervals more often with more training data.
>
> To see that $\eta_n = O(1/\sqrt{n})$, we recall the definition that $\eta_n = 2R_n(\Pi(\mathcal{F})) + M\sqrt{\frac{\ln(1/\delta)}{n}}$ when $R_n(\Pi(\mathcal{F}))$ is the Rademacher complexity. We can show that $R_n(\Pi(\mathcal{F}))$ can be reduced to $R_n(\mathcal{F})$ and it is known that for many hypothesis class such as a class of two-layer neural networks with bounded weights or linear models, $R_n(\mathcal{F})$ is $O(1/\sqrt{n})$. Therefore, we can conclude that $\eta_n = O(1/\sqrt{n})$. We have provided a rigorous derivation of this in Appendix A.
>
> On the effect of the $\eta$ on the final generalization bound, we have provided an additional Theorem in Appendix A that explicitly how the final error bound decay with the number of training sample $n$ (which also affect how large $\eta$ is).

---

> ### Author Response · Authors · 2024-11-25
>
> Dear reviewer,
>
> In the revision, we have comprehensively addressed your main concern regarding explicit bounds. A new Appendix section was added that shows $\eta_n = O(1/\sqrt{n})$ for classes with the Rademacher complexity $R_n = O(1/\sqrt{n})$. In the rebuttal comment before this one, we elaborate on this bound, as well as your other concerns.
>
> We request you to reconsider your score.
>
> Authors

---

> ### Comment · Reviewer_E2Ty · 2024-11-28
>
> I thank the authors for the responses and clarifications on explicit bounds. I am increasing my score to 5.

---

### Author Response · Authors · 2024-11-20
**Summary response to all reviewers with additional theoretical results**

We thank all reviewers for detailed and constructive feedback, and suggestions to improve the paper. Reviewers found our problem setting well-motivated. They appreciated the exposition, finding the paper "well-organized", "clear", and "easy to follow". The key improvement to previous work––a finite-sample generalization analysis under less restrictive assumptions––was clear to all reviewers.

A recurring concern among reviewers was the opacity of our main results, particularly the lack of clarity on how generalization depends on the number of samples $n$. A major update in the revised version is **explicit sample-complexity bounds for classes whose Rademacher complexity is $O(1/\sqrt{n})$**. We show that apart from irreducible additive terms, the part of the excess risk that can be brought down by learning also goes down at the same rate $O(1/\sqrt{n})$. Thus, in the setting we analyze, the statistical behavior of learning with interval feedback is similar to behavior when learning with point feedback. This result and its proof are currently in Appendix A of the main paper, and will also be incorporated into the main paper for the camera-ready version.

We have incorporated additional reviewer suggestions in this revision. These include an alternative definition of the ambiguity degree for a regression setting, clarification of proposed methods, problem setting and the theoretical analysis as well as correction of typos. Updates are highlighted in blue in the revised version. Reviewers also had a number of specific questions that we will answer in individual responses. We believe this substantial revision significantly strengthens the paper and hope the reviewers agree.

---

### Meta-Review · Area_Chair_4dUR · 2024-12-11

**Metareview:**

This paper addresses the challenge of regression with interval targets, where the exact target value is replaced by an interval around the true response. The authors propose two methods: the projection approach, which modifies the regression loss to penalize predictions falling outside the interval, and the minimax objective, which minimizes the worst-case loss within the interval. The paper provides theoretical guarantees for both methods and supports them with empirical experiments on real-world datasets.

Reviewers acknowledged the novelty of the paper and appreciated its theoretical contributions, particularly in tackling the regression problem with interval targets. However, the significance of the theoretical results was questioned, with some reviewers finding them less impactful than expected. For instance, the parameter $\eta$, which depends on Rademacher complexity, is challenging to compute accurately. Additionally, the experimental setup was considered too simplistic, relying on small datasets. As a result, the paper was viewed as borderline, with no reviewer offering strong endorsement.

Given the high competition among ICLR submissions, I lean towards rejection. The authors are encouraged to take into accounts the reviewers' comments and resubmit to another venue.

**Additional Comments On Reviewer Discussion:**

After discussions, the authors have addressed some of the reviewers' concerns. However, the paper has not received strong support overall. There remain questions regarding the significance of the theoretical results and ongoing concerns about the experimental setup.

---

### Decision · Program_Chairs · 2025-01-22

Reject